cognition/psychology/health and disease and epidemiology

coronavirus, COVID-19, optimism bias, pandemic, risk perception

**Authors for correspondence:**
Benjamin J. Kuper-Smith
e-mail: bjks.science@gmail.com
Christoph W. Korn
e-mail: christoph.korn@med.uni-heidelberg.de

# Risk perception and optimism during the early stages of the COVID-19 pandemic

Benjamin J. Kuper-Smith[1,2], Lisa M. Doppelhofer[1,2], Yulia Oganian[3], Gabriela Rosenblau[4,5] and Christoph W. Korn[1,2]

[1]Institute for Systems Neuroscience, University Medical Center Hamburg-Eppendorf, Hamburg, Germany
[2]Section Social Neuroscience, Department of General Psychiatry, Heidelberg University, Heidelberg, Germany
[3]Center for Integrative Neuroscience, University of Tuebingen, Germany
[4]Department of Psychological and Brain Sciences, George Washington University, Washington DC, USA
[5]Autism and Neurodevelopmental Disorders Institute, George Washington University, Washington DC, USA

BJK-S, 0000-0002-2565-222X

Slowing the spread of COVID-19 requires people to actively change their lives by following protective practices, such as physical distancing and disinfecting their hands. Perceptions about the personal risk of COVID-19 may affect compliance with these practices. In this study, we assessed risk perception and optimism about COVID-19 in a multinational (UK, USA and Germany), longitudinal design during the early stages of the pandemic (16 March 2020; 1 April 2020; 20 May 2020). Our main findings are that (i) people showed a comparative optimism bias about getting infected and infecting others, but not for getting severe symptoms, (ii) this optimism bias did not change over time, (iii) optimism bias seemed to relate to perceived level of control over the action, (iv) risk perception was linked to publicly available information about the disorder, (v) people reported adhering closely to protective measures but these measures did not seem to be related to risk perception, and (vi) risk perception was related to questions about stress and anxiety. In additional cross-sectional samples, we replicated our most important findings. Our open and partly preregistered results provide detailed descriptions of risk perceptions and optimistic beliefs during the early stages of the COVID-19 pandemic in three Western countries.

# 1. Introduction

The pandemic of the new Coronavirus COVID-19 requires massive action from governments, industry and citizens to reduce its spread. Best practices, such as minimizing direct physical contact with others (physical distancing) and increased personal hygiene require individuals to actively change their lifestyles [1]. For COVID-19, it is especially important that all citizens follow such guidelines, even those without symptoms, because COVID-19 can be spread by asymptomatic people [2–4]. Ultimately, the success of regulations depends on citizens' compliance.

Individuals' beliefs about their probabilities of contracting and transmitting COVID-19 may determine how much they are willing to change their behaviour. In general, self-related subjective beliefs about future events tend to be optimistically biased [5–8]: people estimate that negative events are less likely to happen to them than to a similar other person, while the opposite is true for positive events. This phenomenon is conceptualized as comparative optimism bias [9,10]. Applied to the current situation, individuals might believe that they are less likely to get infected and to infect others with COVID-19 [11]. Such optimism may also extend to subjective beliefs about following best practice guidelines: if people believe they are not as likely to get infected as other people, they might therefore believe that implementing best practices to minimize the risk of COVID-19 is not as necessary for themselves compared with others. On the other hand, it might lead individuals to believe that they will be more able to comply with best practice guidelines such as physical distancing. In addition, COVID-19 adds another complication for accurately estimating personal risk: especially in the beginning of the pandemic, individuals had no access to definitive statistics due to the novelty of this disease—in contrast to other diseases, such as influenza or sexually transmitted diseases, for which reliable statistics have been long established and publicly available. Moreover, the COVID-19 pandemic evolved rapidly, which changed personal experience as well as publicly available information and public policies.

Multiple lines of research have discussed whether and how optimism can be adaptive or maladaptive for the self [12–15]. Mild optimism can be adaptive. For example, trait optimism predicts physical and mental health (e.g. [16]), possibly via effects related to coping [17]. Relatedly, depressive patients have reduced levels of optimism relative to healthy controls [18,19]. Extreme optimism, however, seems to result in overly high risk taking [20].

Optimism about COVID-19 might have adaptive effects (e.g. protection from detrimental levels of anxiety) or maladaptive consequences (e.g. defiance of regulations and accelerating its spread). This will to some extent depend on what exactly people are optimistic about: if individuals naively believe that they are at a lower risk of contracting or spreading the disease, they may not see the necessity of following best practices around hygiene and physical distancing. In that sense, optimism about COVID-19 might be maladaptive for self and for others because people who follow best practices less strictly might contribute more to the spread of the disease. On the other hand, physical distancing (and quarantine in the extreme case) can be extremely stressful and problematic for mental health [21]. From this point of view, an optimistic belief about one's ability to deal with such a situation might be helpful in following through with physical distancing guidelines, while individuals who think that physical distancing will be very tough for them might be less likely to follow those guidelines. Based on these arguments, optimism for COVID-19 could be adaptive, maladaptive, some combination of the two, or neither.

In this study, we investigated people's risk perceptions and optimism with respect to various measures related to COVID-19, where these perceptions might come from, and whether they predict later adherence to protective measures. To do so, we conducted a longitudinal study during the early stages of the COVID-19 pandemic. Parts of these analyses were preregistered (https://osf.io/89ndm, i.e. in the following sections, we specify *a priori* hypotheses and exploratory analyses). At each time point, we also collected data from additional cross-sectional samples, which we used to test which of the findings reported here replicate. This replication was preregistered after we completed the analyses of the longitudinal dataset (https://osf.io/ukz8n).

# 2. Methods

## 2.1. Overall strategy

We collected two datasets. First, in a within-participants design, data were collected from the same participants at all time points (Sample 1). We used this dataset for the initial longitudinal analyses

| each sample contains DE, UK and US | | |
|---|---|---|
| Sample 1 | Sample 1 | Sample 1 |
| Sample 2 | | |
| | Sample 3 | |
| | | Sample 4 |
| 16.03.2020 | 01.04.2020 | 20.05.2020 |
| T1 | T2 | T3 |

**Figure 1.** A schematic of when we collected data from which samples. In the top row, the blue sample shows our within-participants design. These are the 432 participants who took part three times. The orange samples are independent, new samples that we collected at the same time as the blue samples, but each with a completely new set of participants. The bottom of the figure displays the dates at which the data was collected. The lowest row displays the names we will use to refer to the three time points in the main text. Samples 2–4 each contain new participants from Germany, the UK and the USA. Samples 1 and 2 were split *post hoc* based on whether participants took part in all three times points (in which case they were part of Sample 1, or not (in which case they were part of Sample 2).

reported here. Second, at the same time points, we also collected data from independent participant groups (Samples 2–4), which we will use for replicating our exploratory results. Figure 1 displays which samples were collected when.

## 2.2. Participants

Participants were recruited via www.prolific.ac [22]. We only included participants who completed the entire questionnaire. Further inclusion criteria were:

1) Participants were current residents of the respective countries. We selected UK, US and Germany for the following (rather pragmatic) reasons: First, there were many (more than 1000) active participants available on Prolific. Second, we were able to compile surveys in the participants' native languages quickly enough to start data collection fast. Third, at the time, each country's government had a different approach to dealing with the pandemic.
2) Participants had a prior approval rate of 90–100% on Prolific. When participants participate on Prolific, the experimenters can deny payment to the participant (e.g. if they complete the study too fast to have paid attention, if they miss attention tests). By choosing prior approval rates of 90–100%, we can pre-emptively exclude many unserious participants and thereby increase our data quality.
3) For Sample 1, at T2 and T3, we only invited participants who had taken part in all previous data collections (i.e. T1, and T1 and T2); for Samples 3 and 4, we excluded anyone who had previously taken part in any of our studies.
4) Prolific does not have any participants younger than 18, so this was our imposed lower age limit. We did not set an upper limit.
5) Participants had to take part on a desktop/laptop and were prohibited from taking part on a mobile phone or tablet. This was done to improve data quality, assuming that people sitting at a desktop are not commuting or doing too many other distracting tasks.
6) At the end of the survey, people rated how many problems they had with the survey due to language difficulties. One person reported frequent difficulties and was therefore excluded.

In Sample 1, 432 participants took part (Germany: 135, UK: 206, USA: 91). Participants had a mean age of 33.3 years (s.d. = 11.3; range: 18–81); when asked for their gender, 62.5% selected 'female', 37.0% selected 'male' and 0.5% selected 'other'. In addition to the longitudinal Sample 1, at each time point, we also collected cross-sectional data from each country. While Samples 3 and 4 were intentionally collected as independent replication samples, Sample 2 was initially part of the longitudinal sample (which was reported in the first version of our preprint, see https://psyarxiv.com/epcyb/ Version 1 [23]): not all participants that were initially in Sample 1 responded at all time points, and for those that only participated at T1 and T2 we used the data from T1 as Sample 2. In other words, of the initial 834 participants who took part at T1, 432 (52%) took part at all three time points and constitute Sample 1, and the remaining 402 participants, who did not take part at all three time points, constitute Sample 2. We do not believe that there is any reason to believe that this Sample 2 will differ from the others: we never advertised our study as a longitudinal sample, and we only kept data collection open for two days to achieve a higher precision of timing. For a breakdown of all demographics for each country, see electronic supplementary material, appendix A.

## 2.3. Procedure and questions

Participants saw our study advertised on prolific.ac and were redirected to soscisurvey.de. After consenting to take part in the study, participants filled in the survey. From T1 to T3, we did not exclude any questions, but at T2 and T3 we added several questions towards the end of the survey. Participants were paid £0.85 for 10 min at T1, £1.30 for 15 min at T2 and £1.84 for 20 min at T3.

For a full list of all questions, see electronic supplementary material, appendix B. Participants filled the questionnaire in by selecting the appropriate responses with their cursor. For many questions, the possible answers ranged from 0 to 100 with differently labelled extremes, depending on the question. There was no default option (the slider appeared only when participants clicked on the line). This is important when comparing scores for self to scores for another person, because this way, if there is a difference of 0, this does not mean that participants simply went with the default—instead, answers had to be specifically selected.

Our questions cover the following main topics:

1) Risk perception for self and for an average other person. These are questions about the probability of getting infected with COVID-19, about the probability of infecting others with COVID-19 (if infected oneself), and about the probability of getting mild or severe symptoms (if infected oneself).
2) Control questions for optimism. These included questions about the probability of getting other health issues (getting the flu, getting an STD, breaking a bone), or suffering other health-related negative consequences due to the pandemic (not getting a place at the doctor's/in hospital due to too high demand).
3) Questions about adherence to preventative behaviours, such as physical distancing and hand washing.
4) Questions about mental health (e.g. anticipated suffering due to pandemic, general anxiety).
5) Demographics.
6) General control questions (such as how many infected people they know, how many people deceased due to COVID-19 they know, whether they had symptoms of COVID-19, etc.).
7) Other general questions about the (societal, financial, etc.) consequences of the pandemic.

## 2.4. Testing for optimism bias

For testing comparative optimism, we followed a standard procedure in the field [10] participants separately rated the probability of various events occurring for themselves and for someone similar to them. Optimism scores were always calculated such that a positive number indicates optimism bias (positive events: self-other; negative events: other-self).

To introduce and describe the concept of a 'similar other person', we used similar age, sex and city/area in this study. These variables are key factors with respect to COVID-19: older people [24–26] and men are more at risk from suffering severe symptoms [27], and due to human-to-human transmission, the spread of infected people is not distributed evenly, but in clusters [28]. If the other person is in the same age bracket, has the same sex and is from the same area, we can exclude that any difference is due to perceived differences in those COVID-19-relevant categories. At T1, we only

mentioned age and location in the description of the average person; for T2 and T3, we added biological sex as third factor.

## 2.5. Overall approach and preregistered analyses

Our main analysis is divided into four sections: first, we characterise people's absolute and relative risk perceptions for three questions about COVID-19. Second, we characterize factors that might have influenced these perceptions, such as known risk factors (e.g. age, gender, overall health) and more personal characteristics (e.g. media consumption, overall comparative optimism). Third, we test whether risk perception at one time point predicts self-reported engagement in protective measures at a later time point. For all of these sections, we use the longitudinal Sample 1 only. Fourth, we use the cross-sectional samples 2–4 to test which of the main analyses replicate in independent samples.

Participants were asked three main questions directly related to their risk perception about COVID-19: (i) the probability of getting infected with COVID-19 (hereafter: *Get COVID*), (ii) if infected themselves with COVID-19, the probability of infecting someone else (hereafter: *Infect Others*), and (iii) if infected themselves with COVID-19, the probability of developing severe symptoms that require hospitalization (hereafter: *Severe Symptoms*). While *Severe Symptoms* is a single-item question, the first two questions were asked across different contexts: *Get COVID* was rated separately for four different time horizons (within the next two weeks, within the next two months, within the next year, within your lifetime); *Infect Others* was asked separately for six different social contexts (family, friends, colleagues, strangers during a leisure activity, strangers during vacation, strangers while doing public chores (commuting, buying groceries, etc.)). In the following, when referring to *Get COVID*, we refer to the average rating per participant across the four time horizons, unless specified otherwise; likewise, when referring to *Infect Others* we are referring to an average per participant over the six social contexts, unless specified otherwise.

All three questions were asked for self and for a person similar to the self. When referring to 'absolute risk perception', we refer to the probability of an event happening to oneself; when referring to 'relative risk perception', we refer to the difference in probability of an event happening to someone like you and the probability of that event happening to oneself (i.e. $p_{other}$-$p_{self}$). A positive score for relative risk perception indicates a comparative optimism bias (i.e. the probability for these negative events is rated as higher for the average person than for oneself).

This study includes preregistered analyses (https://osf.io/89ndm) for T2. Specifically, our three preregistered hypotheses were: first, people would show an optimism bias at T2 for the questions *Get COVID* (for the time horizon 'next 2 weeks') and *Infect Others*. This hypothesis rests on the empirical data we collected at T1 and had published in our initial preprint (Version 1 of [23]), and on a similar study by Wise *et al.* [29] that also found a comparative optimism bias for getting infected with COVID-19. Second, these optimism biases would reduce from T1 to T2; again, this hypothesis is based on the previous results by Wise *et al.* [29] who found a reduction of optimism bias in the first week of the pandemic. Third, there would be a negative correlation between these optimism biases at T1 and the reported reduction of physical contacts at T2. This hypothesis was based on our initial empirical observations from T1, during which we found a negative correlation between the optimism bias of *Infect Others*, and the perceived necessity of abiding by best practices, such that the stronger the optimistic bias about infecting other was, the less necessary people believed it was to reduce social contacts. Although the preregistration only explicitly mentioned T2, the same logic can be extended to T3. We therefore also test these preregistered hypotheses for the data from T3. As specified in our preregistration, our cut-off for significance testing was $p < 0.005$. All analyses not explicitly labelled as preregistered hypotheses are treated as exploratory analyses; for these exploratory analyses, we use a cut-off for significance testing of $p < 0.05$. The replication was also preregistered (https://osf.io/ukz8n).

# 3. Results

## 3.1. Absolute and relative risk perception

The indices for *Get COVID* and *Infect Others*, calculated by averaging across time horizons (*Get COVID*) and social contexts (*Infect Others*), and by averaging across all time points (T1–3) for *Get COVID*, *Infect Others* and *Severe Symptoms*, had high internal consistency (Cronbach's alpha for *Get COVID* absolute:

**Table 1.** Correlations between the main variables of risk perception, separately for absolute (self) and relative (other-self) risk perception, calculated by averaging across time horizons (*Get COVID*) and social contexts (*Infect Others*), and by averaging across all time points (T1–3) for *Get COVID*, *Infect Others* and *Severe Symptoms*.

| Rho/p-value | | Get COVID | | Infect Others | | Severe Symptoms | |
| --- | --- | --- | --- | --- | --- | --- | --- |
| | | absolute | relative | absolute | relative | absolute | relative |
| Get COVID | absolute | | | | | | |
| | relative | −0.3784/<0.001 | | | | | |
| Infect Others | absolute | 0.2785/<0.001 | −0.0339/0.4816 | | | | |
| | relative | 0.0339/0.4823 | 0.2947/<0.001 | −0.2666/<0.001 | | | |
| Severe Symptoms | absolute | 0.3116/<0.001 | −0.1027/0.0329 | 0.2231/<0.001 | 0.1633/<0.001 | | |
| | relative | −0.1961/<0.001 | 0.2750/<0.001 | 0.0492/0.3079 | −0.0572/0.2356 | −0.6188/<0.001 | |

0.9183; *Get COVID* relative: 0.8941; *Infect Others* absolute: 0.7939; *Infect Others* relative: 0.7622). The absolute risk perception measures of the three questions correlated positively with the absolute risk perception measures of the other questions, as did the relative risk perception questions, and within each question, there was a negative correlation between absolute and relative risk perception (table 1).

### 3.1.1. Get COVID

Participants in our study believed that there was a substantial risk that they would get infected with COVID-19 (figure 2 shows an overview of the three risk perception questions). The mean score for *Get COVID* averaged across the four time horizons for 'self' was 49% at T1, 46% at T2 and 35% at T3. For the time horizon 'lifetime', the mean score was always above 50%, indicating that participants thought they were more likely than not to get infected with COVID-19 during their lifetimes (note: our data were collected many months before vaccinations for COVID-19 were developed and discussed publicly). While absolute risk perception (averaged across T1–3) increased with longer time horizons ($F_{3,1724} = 149.04$, $p < 0.001$, $\eta_p^2 = 0.1708$), relative risk perception (averaged across T1–3) slightly reduced with longer time horizons ($F_{3,1724} = 8.57$, $p < 0.001$, $\eta_p^2 = 0.0145$; see electronic supplementary material, appendix C).

For relative risk perception, a clear picture emerged: for all time horizons across all time points, the mean response for 'self' was always lower than for 'other', suggesting an overall optimism bias for *Get COVID*, such that people estimated this negative event to be more likely to happen to someone else than to themselves. We therefore found evidence for the first part or our first preregistered hypothesis: risk perception for 'self' was statistically significantly lower than that for 'other' at T2 for the time horizon 'next 2 weeks' ($t_{431} = -11.69$, $p < 0.001$, $d = -0.56$; the preregistered hypothesis was specifically about the time horizon 'next 2 weeks', rather than the average of all four time horizons).

To test whether risk perceptions and comparative optimism bias changed during the early stages of the pandemic for *Get COVID*, we ran a 2 (person: self/other) ∗ 3 (time point: T1–3) repeated-measures ANOVA. We found a significant main effect of person ($F_{1,862} = 177.00$, $p < 0.001$, $\eta_p^2 = 0.3046$), which suggests an optimism bias, and a significant main effect of time point ($F_{2,862} = 109.71$, $p < 0.001$, $\eta_p^2 = 0.6438$), which suggests that risk perception decreased over time. The interaction between person and time ($F_{2,862} = 3.07$, $p = 0.0471$, $\eta_p^2 = 0.0071$) was only significant at the $p = 0.05$ level though and would not be significant after adjusting for multiple comparisons; additionally, the effect size of this interaction was small. Testing our second preregistered hypothesis, we did not find a statistically

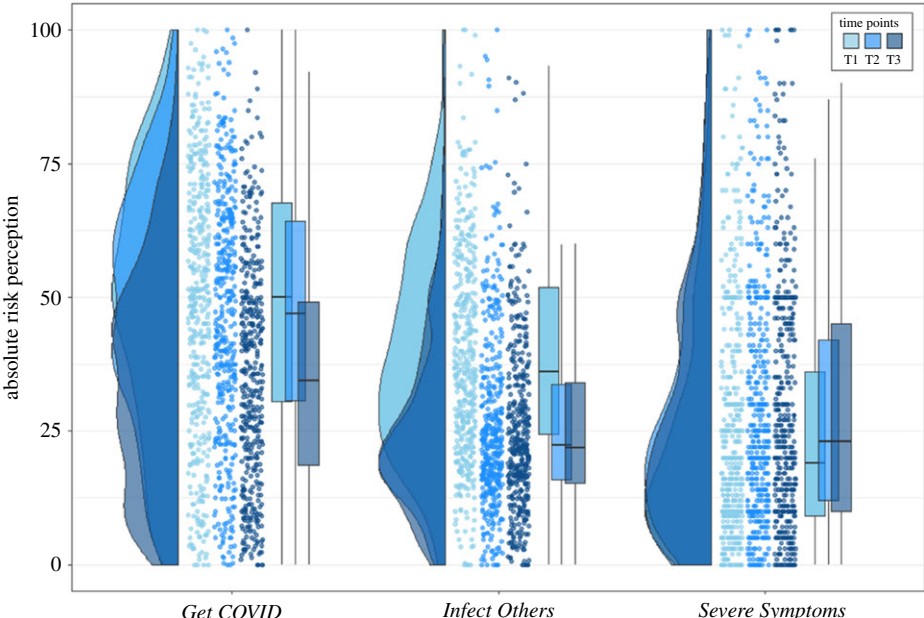

**Figure 2.** Raincloud plots for absolute risk perception, separately for the three COVID-19-related risk perception questions, separately for each time point. Each dot represents the data from one participant of Sample 1. For *Get COVID*, we averaged across the four different time horizons (next two weeks, next two months, next year, lifetime) and for infect others, we averaged across the six social contexts (family, friends, colleagues, strangers during leisure activities, strangers during vacation, strangers during daily chores like commuting and grocery shopping); severe symptoms is a single-item questions. All raincloud plots in this paper were based on [30].

significant reduction between T1 and T2 for the optimism bias scores for the time horizon 'next 2 weeks' ($t_{431} = -1.55$, $p = 0.122$, $d = -0.075$).

Taken together, this suggests that although absolute risk perception reduces over time, there does not seem to be a substantial change in people's optimism bias for *Get COVID* during the early stages of the pandemic. Figure 3 illustrates these findings for all three optimism bias scores for all three time points.

### 3.1.2. Infect Others

Participants believed that there was a substantial risk that they would infect someone else with COVID-19 (if they themselves were infected): the mean risk for *Infect Others* averaged across all social contexts was always estimated to be above 20% (T1: 39%; T2: 26%; T3: 25%). There were significant differences between the different social contexts for *Infect Others* for both absolute risk perception (averaged across T1–3; $F_{5,2586} = 218.45$, $p < 0.001$, $\eta_p^2 = 0.2290$), and for relative risk perception (averaged across T1–3; $F_{5,2586} = 28.1$, $p < 0.001$, $\eta_p^2 = 0.0490$; see electronic supplementary material, appendix C).

For the relative risk perception of *Infect Others* a clear picture emerged (figure 2): for all social contexts across all time points, the mean response for 'self' was always lower than for 'other', suggesting an overall optimism bias for *Infect Others*. We thus found evidence for the second part of our first preregistered hypothesis: the risk perception for 'self' was statistically significantly lower than that for 'other' at T2 for the average across all six social contexts for *Infect Others* too ($t_{431} = -21.99$, $p < 0.001$, $d = -1.058$).

To test whether risk perceptions and comparative optimism bias changed during the early stages of the pandemic for *Infect Others*, we ran a 2 (person: self/other) * 3 (time: T1–3) ANOVA. There was a main effect of person ($F_{1,862} = 806.23$, $p < 0.001$, $\eta_p^2 = 0.7150$), which suggests an optimism bias, a main effect of time ($F_{2,862} = 139.66$, $p < 0.001$, $\eta_p^2 = 0.5870$), which suggests that risk perception decreases over time, and an interaction between person and time ($F_{2,862} = 4.54$, $p = 0.0109$, $\eta_p^2 = 0.0104$). As for *Get COVID*, this interaction effect for *Infect Others* was only significant at the $p = 0.05$ level and would not be significant if corrected for multiple comparisons. Additionally, the effect size of this interaction was small.

Taken together, this suggests that although absolute risk perception decreased over time, there was no substantial change in people's optimism bias for *Infect Others* during the first two months of the pandemic. Testing our second preregistered hypothesis (which specified comparing T1 and T2 only,

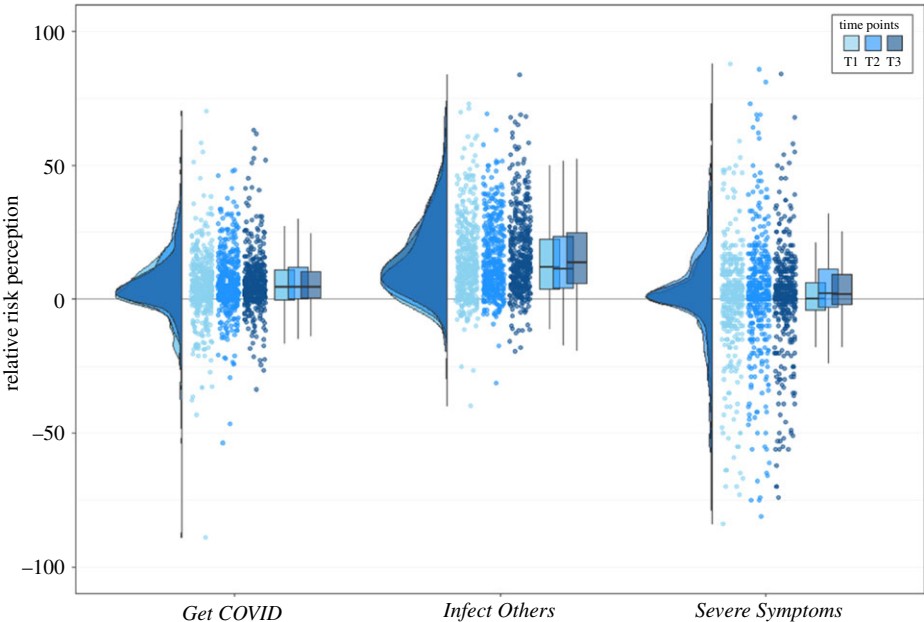

**Figure 3.** Raincloud plots for relative risk perception (optimism bias; other-self), separately for the three COVID-19-related risk perception questions, separately for each time point. Each dot represents the data from one participant of Sample 1. For *Get COVID*, we averaged across the four different time horizons (next two weeks, next two months, next year, lifetime) and for *Infect Others*, we averaged across the six social contexts (family, friends, colleagues, strangers during leisure activities, strangers during vacation, strangers during daily chores like commuting and grocery shopping); *Severe Symptoms* is a single-item question.

with a repeated-measures $t$-test), we did not find a statistically significant reduction between T1 and T2 for the optimism bias scores for the *Infect Others* ($t_{431} = -0.11$, $p = 0.9118$, $d = -0.005$).

### 3.1.3. Severe Symptoms

As with *Get COVID* and *Infect Others*, we ran a 2 (person: self/other) $*$ 3 (time: T1–3) ANOVA, to test for risk perceptions and comparative optimism bias change during the early stages of the pandemic for *Severe Symptoms* (figure 2). There was no main effect of person ($F_{1,862} = 1.45$, $p = 0.2299$, $\eta_p^2 = 0.0062$), which suggests no optimism bias, a main effect of time ($F_{2,862} = 17.85$, $p < 0.001$, $\eta_p^2 = 0.1163$), which suggests that risk perception increased over time, and no significant interaction between person and time ($F_{2,862} = 2.32$, $p = 0.0993$, $\eta_p^2 = 0.0053$). Taken together, this suggests that risk perception for *Severe Symptoms* differed from risk perception for *Get COVID* or *Infect Others*: while the other two showed strong optimism biases and decreasing absolute risk perception, *Severe Symptoms* showed no optimism bias, and absolute risk perception seemed to increase slightly over time (significant, but with small effect size).

### 3.1.4. Comparing optimism scores for COVID-19-related questions, other diseases and proximity

When comparing optimism scores across the three optimism questions (*Get COVID*, *Infect Others*, *Severe Symptoms*), we found significant differences. Given that optimism scores did not change over time for any of the three questions, we calculated the mean optimism score per participant per question over the three time points, and compared this aggregate optimism score across questions. A one-way repeated-measures ANOVA revealed an overall significant effect of question (*Get COVID*, *Infect Others*, *Severe Symptoms*; $F_{1,862} = 173.38$, $p < 0.001$, $\eta_p^2 = 0.2869$). *Post hoc* tests revealed a significantly higher optimism bias for *Infect Others* than for *Get COVID* (mean difference: 9.7551 (95% CI: 7.8980-11.6121); $p < 0.001$) and *Severe Symptoms* (mean difference: 14.4644 (95% CI: 12.6073–16.3214); $p < 0.001$), and a significantly higher optimism bias for *Get COVID* than for *Severe Symptoms* (mean difference: 4.7093 (95% CI: 2.8522–6.5664); $p < 0.001$).

One interpretation of these differences is that optimism bias is affected by the perceived sense of control over the outcome. At the time of data collection, there was no known cure for COVID-19, so the probability of experiencing severe symptoms depended mostly on a person's immune system,

something that depends to a considerable degree on genetics and other biological factors, and is thus largely outside of one's control. For getting infected, however, there is quite a lot one can do to prevent this outcome, such as staying home as much as possible, washing hands, maintaining distance and wearing masks. Despite all this, one cannot easily guarantee to not get infected unless one quarantines oneself for an undetermined time. Arguably, one should have the most control over one's actions for *Infect Others* (at least if one is clearly positive for COVID-19): as long as a person abides perfectly by the health guidelines and quarantines themself, one can all but guarantee that no one else will get infected. Based on this reasoning, we ran the following analyses to test the interpretation that a sense of control affects the degree to which people are optimistically biased:

First, in an additional set of risk perception questions about other diseases (probability of getting influenza/getting an STD/breaking a bone), there were significant differences between the optimism biases of these questions ($F_{1,862} = 198.65$, $p < 0.001$, $\eta_p^2 = 0.3155$). *Post hoc* tests revealed that people reported a lower optimism bias for getting influenza than for getting an STD (mean difference: 10.6682 (95% CI: 9.3395–11.9969); $p < 0.001$) and for breaking a bone (mean difference: 2.1082 (95% CI: 0.7796–3.4369); $p = 0.006$), and a significantly lower optimism bias for breaking a bone than for getting and STD (mean difference: 8.5600 (95% CI: 7.2313–9.8887); $p < 0.001$; see electronic supplementary material, appendix D). Although we did not ask participants for their perceived sense of control over these items, it seems reasonable that people feel the most control over whether they will get an STD, followed by whether they will break a bone, and whether they will contract influenza. Thus, the control optimism questions seem to align with the COVID-19 optimism questions in terms of perceived sense of control.

Second, it seems that more proximal (and therefore, presumably, controllable) aspects show a larger optimism bias than more distal ones: within the sub-items of *Get COVID*, the shorter the time horizon, the larger the optimism bias ($F_{3,1724} = 8.57$, $p < 0.001$, $\eta_p^2 = 0.0147$). We also asked participants, if they were to get infected, to what extent different people had done all they could to prevent their infection from happening. We varied who these people were, from the participant themselves to friends and family members, employers, local authorities and the national government. Again, we find that the more proximal someone is the stronger the optimism bias is: comparing the different people, we find that the closer someone is to the participant, the larger the optimism bias is ($F_{4,2155} = 35.69$, $p < 0.001$, $\eta_p^2 = 0.0621$; the mean optimism bias for each item increases from the person themselves to government, with most individual comparisons statistically significantly different; see electronic supplementary material, appendix D).

Third, for *Infect Others*, the sub-item Family has the lowest optimism bias (one-way repeated measures ANOVA between social contexts: $F_{5,2786} = 28.1$, $p < 0.001$, $\eta_p^2 = 0.0515$; *post hoc* comparisons showed family as having the smallest optimism bias; for all comparisons with family $p < 0.001$) and due to many of our participants living with their family (74% of participants report living with children, partner and/or parents), one can control to a much lesser degree whether one will interact with one's family relative to the other sub-items. Taken together, the findings from these questions suggest that optimism bias might relate to perceived control.

## 3.2. Potential influences on risk perception

Next, we investigated the potential influences of risk perception. As described above, there are many different reasons why people might show a comparative optimism. In this section, we analyse what might explain these perceptions. To keep the number of tests tractable, we do not report changes over time here; we therefore analyse risk perception averaged over the three time points, each for absolute and relative risk perception.

### 3.2.1. Personal risk factors

At the time of data collection, several risk factors for COVID-19 were already known, including age, gender, overall health and geographical location [26]. Being older, male and in poor health was known to be associated with higher probabilities of developing severe symptoms once infected.

**Age:** At the time of data collection, it was widely reported in the media that being older was a known risk factor for developing severe symptoms and dying from a COVID-19 infection [24,25]. To test the effect of age on perceived risk, we analysed pairwise correlations between age and absolute and relative risk scores in each of the three main risk perception questions. For *Get COVID*, there was no evidence for a relationship between age and absolute risk perception ($r = -0.0393$, $p = 0.4157$; see

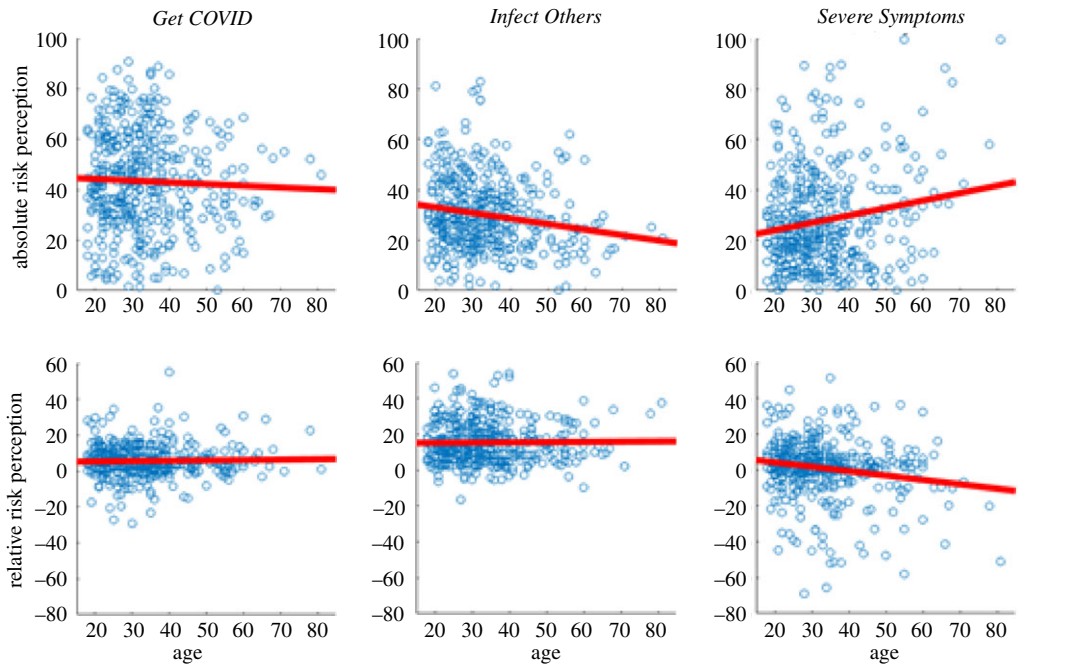

**Figure 4.** Scatter plots for age, and absolute and relative risk perception for the three COVID-19-related risk perception questions. The red line displays the least squares line. For *Get COVID* and *Infect Others*, we averaged across each sub-item (severe symptoms was a single-item question). For each question, we averaged across the three time points (T1–3). *Infect Others* absolute risk and *Severe Symptoms* both absolute and relative risk are statistically significant; all others are not.

figure 4 for scatter plots to all results mentioned in this section) or age and relative risk perception ($r = 0.0274$, $p = 0.5696$). For *Infect Others*, older participants perceived less absolute risk ($r = -0.1776$, $p < 0.001$), but there was no evidence for a relationship between age and relative risk perception ($r = 0.0130$, $p = 0.7875$). For *Severe Symptoms*, older participants had higher scores of absolute risk ($r = 0.1731$, $p < 0.001$) and lower scores of relative risk ($r = -0.1723$, $p < 0.001$). Thus, our participants seemed to have particularly incorporated the information that older people are more likely to suffer from severe symptoms once infected. Although relative risk perception is relative to someone of the same age, older people seem to show a reduced optimism bias with respect to getting severe symptoms.

**Gender:** At the time of data collection, it was reported that men suffered from higher rates of severe symptoms and death due to COVID-19 than women. To test the effect of gender on risk perception, we used Welch's $t$-test ($t_{\text{Welch}}$), due to the different number of men and women in our sample. Furthermore, only two participants selected 'other' as their gender, such that there weren't enough people from this group to run proper analyses, and we had to exclude them.

For *Get COVID*, men ($M = 38.2531$, s.d. $= 19.4670$) showed lower absolute risk perception than women ($M = 46.6901$, s.d. $= 18.6814$; $t_{\text{Welch}}(323.0304) = 4.4094$, $p < 0.001$, $d = 0.445$; see figures 5 and 6 for plots to all results mentioned in this section). This was not the case, however, for relative risk perception ($t_{\text{Welch}}(388.2063) = -0.4095$, $p = 0.6824$, $d = -0.039$), indicating a baseline shift: women reported generally higher risk perception, irrespective of whether it concerns themselves or someone else. For *Infect Others*, absolute risk perception did not differ between men and women ($t_{\text{Welch}}(308.6878) = 0.8142$, $p = 0.4161$, $d = 0.083$), but women ($M = 16.2932$, s.d. $= 11.6474$) had larger relative risk perception than men ($M = 13.9701$, s.d. $= 10.4912$; $t_{\text{Welch}}(361.9566) = 2.1292$, $p = 0.0339$, $d = 0.207$). For *Severe Symptoms*, women had higher perceived absolute risk perception ($M = 29.4173$, s.d. $= 19.9943$) than men ($M = 25.0563$, s.d. $= 17.9717$; $t_{\text{Welch}}(362.5108) = 2.3313$, $p = 0.0203$, $d = 0.226$), but there was no difference in relative risk perception ($t_{\text{Welch}}(384.0029) = -0.9032$, $p = 0.3670$, $d = -0.086$).

**Overall health:** People with (some types of) pre-existing health conditions were at higher risk of getting severe symptoms if infected with COVID-19. We thus asked participants to rate their overall health from 1 (very poor) to 5 (very good). The responses were highly unevenly distributed (very poor: 7, poor: 24, OK: 109, good: 205, very good: 87), so we used a Kruskal–Wallis test after excluding the small fraction of participants who reported very poor health.

For *Get COVID*, there were no significant differences between people with different levels of health (absolute: $H(3) = 7.78$, $p = 0.0507$, $\eta^2_H = 0.0114$; relative: $H(3) = 5.20$, $p = 0.1575$, $\eta^2_H = 0.0052$; see

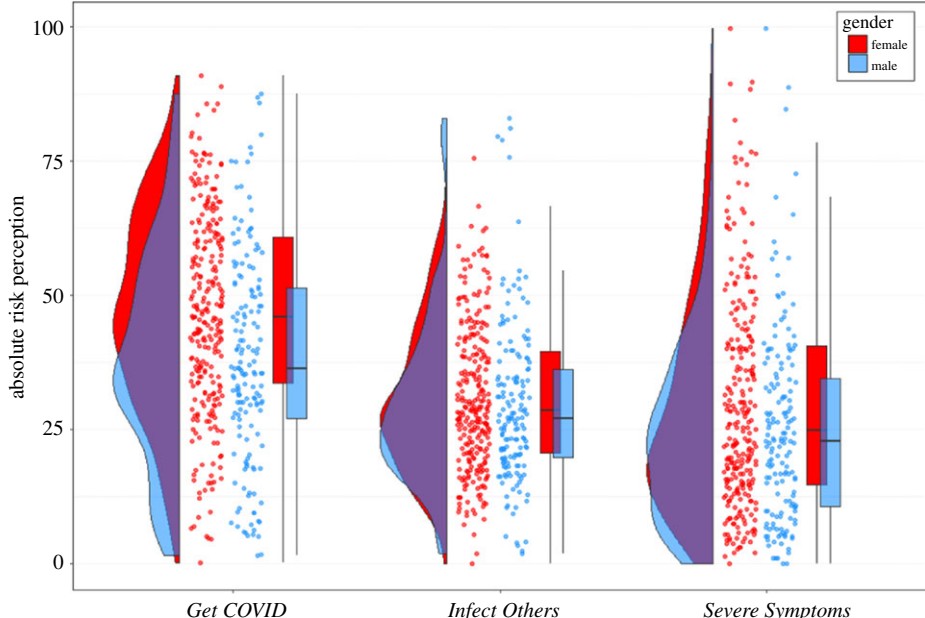

**Figure 5.** Absolute risk perception for the three COVID-19-related risk questions, separately for women (red) and men (blue); only two people selected 'other' and were excluded from this analysis due to limited sample size. For *Get COVID* and *Infect Others*, we averaged across each sub-item (*Severe Symptoms* was a single-item question). For each question, we averaged across the three time points (T1–3).

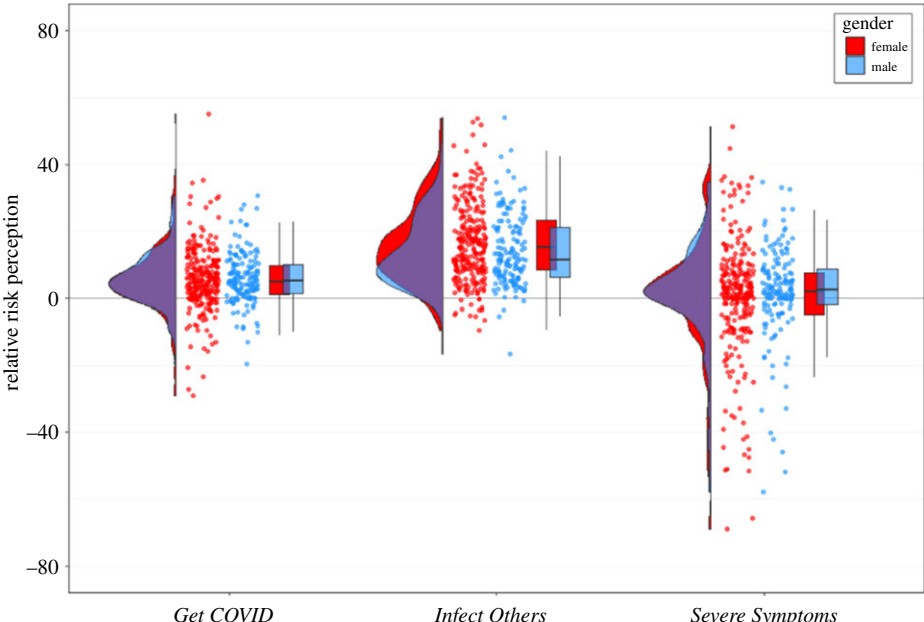

**Figure 6.** Relative risk perception for the three COVID-19-related risk questions, separately for women (red) and men (blue); only two people selected 'other' and were excluded from this analysis due to limited sample size. For *Get COVID* and *Infect Others*, we averaged across each sub-item (*Severe Symptoms* was a single-item question). For each question, we averaged across the three time points (T1–3).

figures 7 and 8 for plots to all results mentioned in this section). For *Infect Others*, there were no significant differences for absolute risk perception (H(3) = 4.30, $p = 0.2308$, $\eta_H^2 = 0.0031$), but for relative risk perception, there were significant differences (H(3) = 9.86, $p = 0.0198$, $\eta_H^2 = 0.0163$). *Post hoc* tests revealed that people with poor health reported significantly higher optimism bias of infecting someone else than people with good ($p = 0.0184$) or very good ($p = 0.0293$) health.

For severe symptoms, there was a negative relationship between health and absolute risk perception: the worse someone's health was, the higher they rated their probability of experiencing severe symptoms if infected with COVID-19 (H(3) = 78.54, $p < 0.001$, $\eta_H^2 = 0.1794$), with a decrease from one level of health

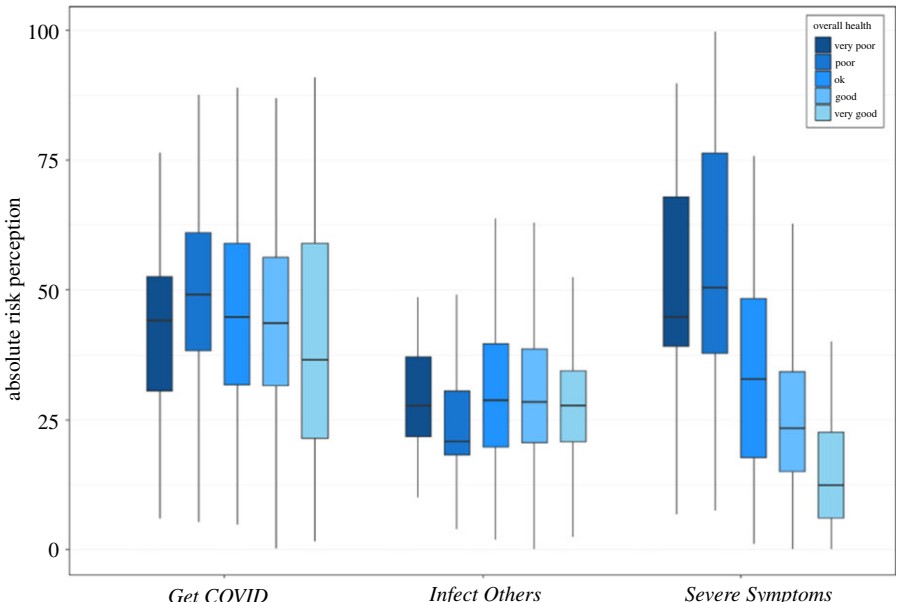

**Figure 7.** Absolute risk perception for the three COVID-19-related risk questions, separately for the five response options for overall health from very poor (darkest blue) to very good (lightest blue). Note that only seven people selected very poor and were excluded from the main analysis; but we represent them here for completeness. For *Get COVID* and *Infect Others*, we averaged across each sub-item (*Severe Symptoms* was a single-item question). For each question, we averaged across the three time points (T1–3). Box plots display the minimum and maximum (end of the lines), the first and third quartile (end of boxes), and the median (horizontal line within the box).

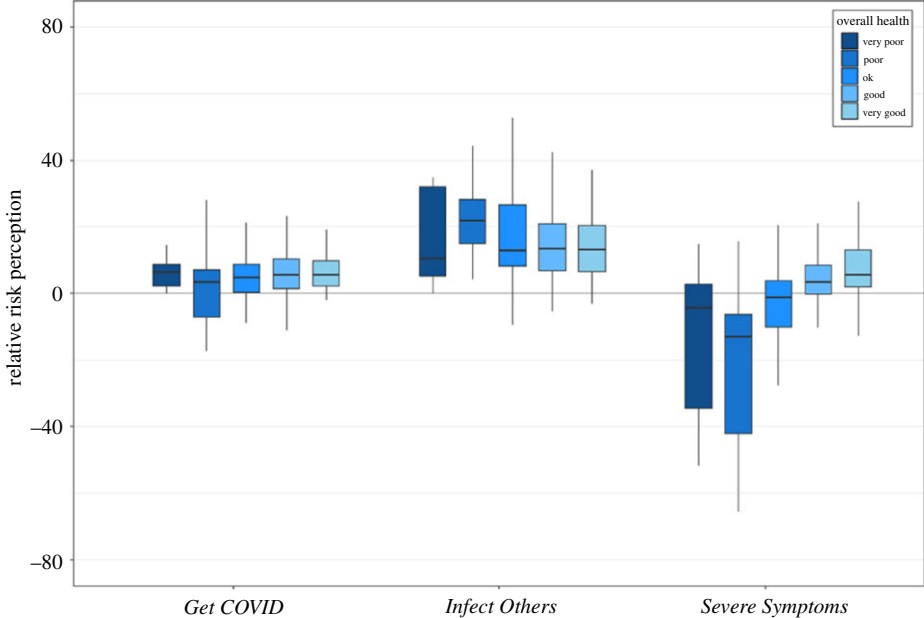

**Figure 8.** Relative risk perception for the three COVID-19-related risk questions, separately for the five response options for overall health from very poor (darkest blue) to very good (lightest blue). Note that only seven people selected very poor and were excluded from the main analysis; but we represent them here for completeness. For *Get COVID* and *Infect Others*, we averaged across each sub-item (*Severe Symptoms* was a single-item question). For each question, we averaged across the three time points (T1–3). Box plots display the minimum and maximum (end of the lines), the first and third quartile (end of boxes), and the median (horizontal line within the box).

to the next (poor/OK $p = 0.0207$, OK/good $p = 0.0073$, all other comparisons $p < 0.001$). For relative risk perception, the reverse was true: the better someone's health, the higher their optimism bias (H(3) = 81.46, $p < 0.001$, $\eta_H^2 = 0.1864$), with a monotonic increase from one level to the next (poor/OK

$p = 0.0093$, good/very good $p = 0.0116$, all other comparisons $p < 0.001$). Thus it seems as if our participants were highly aware of the relationship between pre-existing health conditions and a higher probability of getting severe symptoms once infected with COVID-19.

### 3.2.2. Country

The participants in our study came from three different countries. Due to the different number of participants in each country (DE: 135, UK: 206, USA: 91), we used a Kruskal–Wallis test to determine whether there were any significant differences between countries for absolute and relative risk perception for the three COVID-19-related risk perception questions. There were no significant differences between countries for all absolute and relative risk perception scores (all $p$-values $> 0.06$), with the exception of relative risk perception for *Infect Others* (H(2) = 9.5, $p = 0.0087$, $\eta_H^2 = 0.0175$). *Post hoc* tests revealed that participants from the USA had significantly higher optimism bias scores than participants from the UK ($p = 0.0085$) and Germany ($p = 0.0287$), but that there was no difference between Germany and the UK ($p = 0.9686$). Although there was an overall significant difference for relative risk perception for *Infect Others*, this effect has a small effect size and would not be significant if controlling for multiple comparisons ($p_{\text{Bonferroni}} = 0.005/6 = 0.0083 < 0.0087$). It does thus not seem as if there are any large and meaningful differences in risk perception between our samples in the UK, the USA and Germany. For a table of results and for figures for risk perception in each country see electronic supplementary material, appendix E.

We also asked participants how much they trusted their respective governments to respond adequately to the pandemic. Comparing the scores for each country (averaged across the three time points) revealed overall differences between the countries (H(2) = 89.35, $p < 0.001$, $\eta_H^2 = 0.2036$), and *post hoc* tests revealed that participants from Germany had higher trust in their government than participants from the UK and from the USA (both $p < 0.001$), while people from the UK had higher trust in their government than people from the USA did ($p = 0.0128$). We also asked the same question for trust in science, with similar results: an overall test revealed significant differences between countries (H(2) = 31.34, $p < 0.001$, $\eta_H^2 = 0.0684$), and *post hoc* tests again revealed that people from Germany had higher trust in science than people from the UK ($p < 0.001$) and the USA ($p = 0.0070$), but there were no differences between people from the UK and the USA ($p = 0.2213$). Trust in government (across all countries) was not related to *Get COVID*, neither for absolute ($r = -0.0657$, $p = 0.1726$) or relative ($r = 0.0742$, $p = 0.1234$). While for *Infect Others* there were no effects for absolute risk perception ($r = 0.0249$, $p = 0.6056$), there was a small but significant effect for relative risk perception ($r = -0.1695$, $p < 0.001$). For *Severe Symptoms*, both risk perception measures showed a small but significant correlation, negative for absolute risk perception ($r = -0.0974$, $p = 0.0430$), and positive for relative risk perception ($r = 0.1026$, $p = 0.0330$), although neither effect would be statistically significant after correcting for multiple comparisons using Bonferroni correction. See electronic supplementary material, appendix E for figures.

### 3.2.3. Proximity to infections and deaths

Another factor that might affect risk perception is the personal proximity to people who have been infected or who have died from COVID-19. Although our question allowed people to say how many people they knew who had been infected or had died from COVID-19, about half of people did not know a single person who had been infected or who had died. Thus, for the analysis, we made these variables binary (no one known versus at least one person known). Because the number of people who knew at least one person who was infected or had died increased over time, we only used the data from T3 for analyses regarding proximity (see electronic supplementary material, appendix F for details).

For *Get COVID*, those who knew at least one person who had died or been infected with COVID-19 showed higher absolute risk perception than those who knew no one who had been infected (see electronic supplementary material, appendix G for details for this section). While this pattern was evident for all items for *Get COVID* (infections known directly, infections known indirectly, deaths known), for *Infect Others*, this pattern was only evident for infections known directly (no significant differences for the other two items). For relative risk perception, for both *Get COVID* and *Infect Others* only the item infections known indirectly showed a significant effect: those who indirectly knew at least one person who had been infected with COVID-19 showed lower relative risk perception than those who knew no one indirectly who had been infected. For *Severe Symptoms*, there were no

significant differences between those participants who knew at least one person and those participants who knew no one who had been infected or died.

### 3.2.4. General optimism bias

Another factor that might affect people's risk perception, especially their comparative optimism bias, is their personalized general optimism bias for other diseases. Put differently, people might have a certain baseline optimism bias, and this might in turn affect how they perceive risks around COVID-19.

To test for this possibility, our survey included three questions related to getting health problems other than COVID-19: contracting the flu, getting a sexually transmitted disease (STD), and breaking a bone (see electronic supplementary material, appendix H for statistical details; these are the same questions reported in §3.1.4: Comparing optimism scores for COVID-19-related questions, other diseases and proximity). As with *Get COVID*, we asked this question for four different time horizons (next two weeks, next two months, next year, lifetime). For the analysis in this section, we calculated the average across all four time horizons for each question, and then calculated the average across the three questions, to reach an average optimism score for non-COVID-19 health issues. We then correlated this overall score with the absolute and relative risk perception scores for our three questions. The overall optimism score was not related to absolute risk perception for the COVID-19 risk perception questions, but showed a significant and positive correlation with relative risk perception. In other words, people seem to have an overall optimism bias profile that applies to COVID-19-related questions.

### 3.2.5. Media consumption

A final potential influence on risk perception that we assessed is how much media people consume about COVID-19. For *Get COVID*, absolute risk perception increased with more media consumption ($r = 0.1384$, $p = 0.0040$), but there was no relationship between media consumption and relative risk perception ($r = 0.0556$, $p = 0.2492$). For *Infect Others*, there were no significant relationships between risk perception and media consumption (absolute risk perception: $r = 0.0301$, $p = 0.5325$; relative risk perception: $r = 0.0276$, $p = 0.5670$). For *Severe Symptoms*, absolute risk perception increased with more media consumption ($r = 0.1362$, $p = 0.0046$), but relative risk perception decreased with more media consumption ($r = -0.1030$, $p = 0.0324$). It should be added that for media consumption the direction of causality could go either way: people might believe there to be a higher risk of getting infected because they consume more media, or they might be at an objectively higher risk of getting infected, and therefore follow media reports on the topic more closely. We did not collect any data about what kind of media (e.g. news or social media) people consumed, so our data does not provide any information about how different types of media might relate to risk perception.

## 3.3. Potential consequences of risk perception

Having described risk perception and potential influences, we now turn to the consequences of risk perception. What does risk perception (potentially) lead to? Initially, we intended to predict adherence to protective measures (physical distancing and hand washing) from risk perception. We preregistered the hypothesis that optimism bias at T1 for *Get COVID* and *Infect Others* would correlate negatively with hygiene measures and the reduction of physical contacts at T2. As with the previous sections, we also planned on expanding this hypothesis to T3 (i.e. correlating risk perception at T2 with protective measures at T3). Unexpected ceiling/floor effects, however, precluded us from running these analyses appropriately (see below): our participants reported almost complete adherence to protective measures such that the variance in the data is not sufficient for a meaningful correlation.

### 3.3.1. Reduction of physical contacts

In our preregistration, we hypothesized that risk perception at T1 would negatively predict adherence to protective measures at T2. To test this, we specified to correlate optimism bias at T1 for *Get COVID* and *Infect Others* with the subjective reduction of physical contacts since the beginning of the pandemic. During data analysis, however, we realized that without an objective baseline, it is unclear what this difference actually measures: if someone does not report a reduction of physical contacts, is this due to them not wanting to, or due to them not being able to reduce contacts (for example because they have to care for their elderly parents, or because they already have no contacts)? Thus, we added a

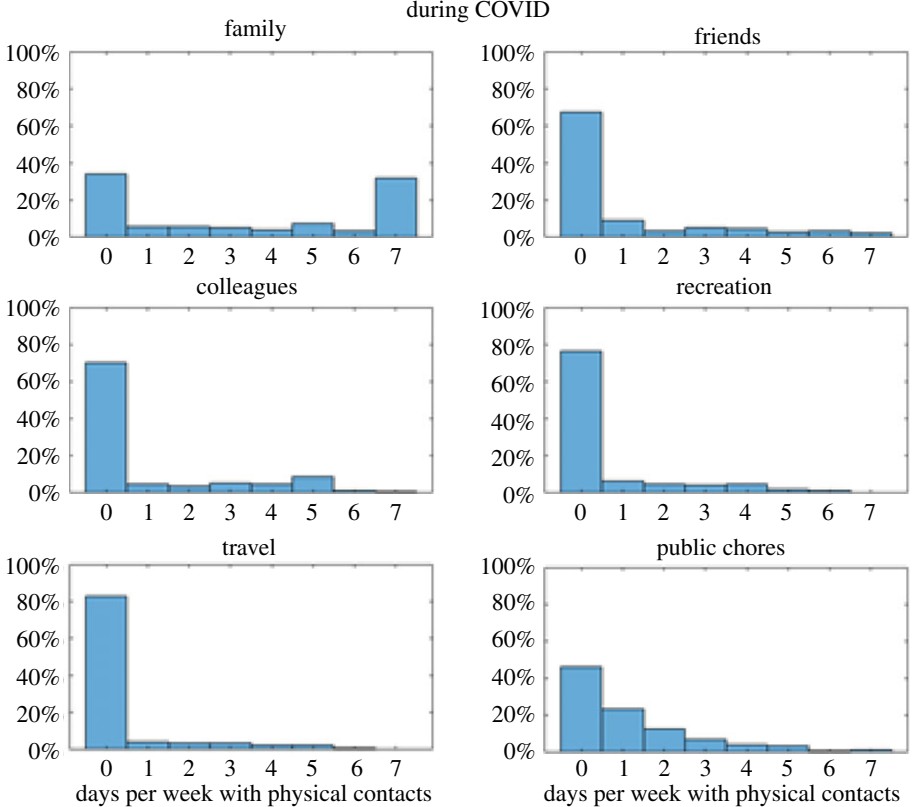

**Figure 9.** Proportions of the number of days per week during which participants had physical contacts during COVID-19, rated separately for the six social contexts. All questions here are from T2, but those for T3 are very similar.

different question at T2 and used this one instead to assess the reduction of physical contacts: participants reported on how many days per week they had physical contacts with people, for their usual life before the pandemic and for their lives during the pandemic, both times separately for the six social contexts (family, friends, colleagues, strangers during leisure activities, strangers during travel, strangers doing chores such as commuting and grocery shopping). This provides a baseline (pre-COVID-19), the number of contacts during the pandemic, and their difference. In the following, we use this question to assess reduction of physical contacts during the pandemic.

At T2 (almost identical for T3), the vast majority of participants reported zero days with physical contacts for the contexts 'friends' (68%), 'colleagues' (70%), 'recreation' (76%) and 'travel' (82%). For public chores (which includes commuting and grocery shopping), the distribution is also skewed, albeit less than the other four contexts mentioned above (70% report zero or one day with physical contacts). The context 'family' shows a bimodal distribution, such that around 70% of people report either zero physical contacts (34%) or daily physical contacts (32%) with their family members Thus, almost all participants reported reducing their physical contacts as far as possible (see figure 9 for the number of contacts during COVID, and see electronic supplementary material, appendix I for figures before COVID).

Owing to these floor effects, the difference (Contacts$_{Normal}$-Contacts$_{COVID}$) is almost identical to the pre-COVID-19 numbers (because for most participants and most contexts, zero is subtracted). Thus, the amount of contacts pre-COVID-19 and the reduction of contacts during COVID-19 (i.e. the difference) show very high correlations (Rs of approx. 0.7–0.8 for friends, colleagues, recreation and travel). Thus, taking the difference between pre-COVID-19 and during COVID-19 does not measure the reduction of physical contacts in a meaningful way. Almost everyone in our sample reported to have reduced as much as they could. While this might be good for society (and was in part due to governmental restrictions), this lack of variability in physical reductions precluded us from assessing the relationship between risk perception and reduction of physical contacts with change scores.

We instead opted for using regressions in which we tried to predict the number of contacts during COVID (measured at T2) from risk perception (measured at T1), controlling for number of physical contacts before COVID. We ran separate regressions for each social context, such that we tried to

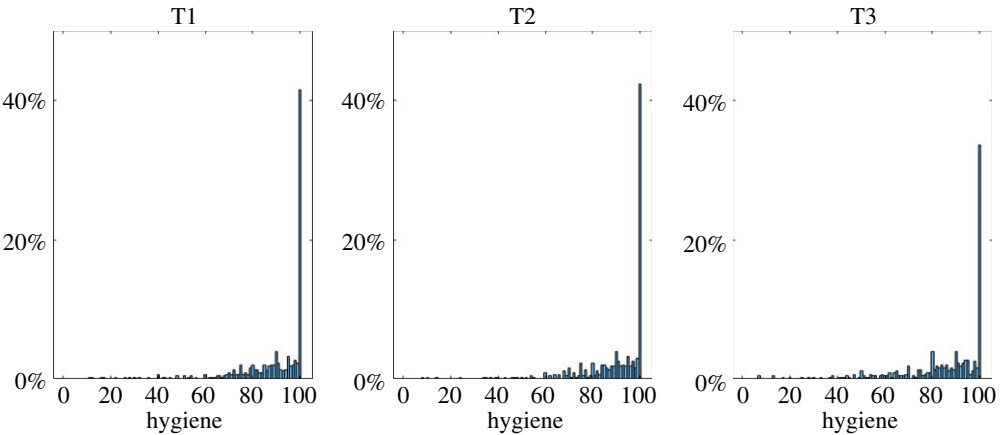

**Figure 10.** Proportions of how much people abide by the hygiene recommendations, separately for each time point. The slight drop at T3 is probably due to us adding 'wearing face masks' as an example, rather than people sticking less to the previous examples of hand washing and disinfecting.

predict physical contacts in each social context (family, friends, colleagues, recreation, travel, public chores, and the average of those six social contexts).

Overall, results were similar across all social contexts: risk perception did not predict the number of physical contacts, but the number of physical contacts before COVID was always a significant predictor. Table 2 shows the results for predicting physical contacts at T2 from risk perception at T1 and we repeated the same analysis for predicting physical contacts at T3 from risk perception at T2, with similar results: the number of physical contacts before COVID was always a significant predictor, but for the context Family, the significant risk perception predictors from table 2 were no longer significant. For some of the other contexts, in the T2/T3 analysis a single risk perception measure was a significant predictor, but not in any systematic way, and only at the level of $p = 0.05$, not correcting for multiple comparisons. Thus, in our sample, it does not seem as if risk perception is a significant predictor of the number of physical contacts during the pandemic.

### 3.3.2. Hygiene behaviour

Similar to the reduction of physical contacts, we originally planned to correlate risk perception with adherence to hygiene recommendations. But the item *Hygiene* is also heavily skewed, such that at T2 42% of participants selected the maximum score (100 out of 100), and 55% selected a score of at least 95 (figure 10). Given that there still are many people who did not select the maximum (or close to the maximum), we grouped participants: those who claim to abide entirely or almost entirely by the hygiene recommendations (greater than or equal to 95%), and everyone else (less than 95%). Instead of correlating hygiene with risk perception, we compared those two groups, testing whether there were differences in risk perception at T1/2 for those who report ≥95%/<95% adherence to hygiene recommendations at T2/3.

Comparing absolute and relative risk perceptions, only one test was statistically significant at the $p = 0.005$ level as preregistered, but this would no longer be significant when correcting for multiple comparisons (see electronic supplementary material, appendix J for the results to all tests).

### 3.3.3. Mental health

At T2 and T3, we also asked people one question about their overall stress and anxiety, and another question about their financial worries and/or losing their jobs. As with most other questions, we asked participants to rate these questions both for themselves and for someone like them, i.e. how much stress and anxiety/financial worry they felt themselves, and how much stress and anxiety/financial worry someone similar to themselves felt. Overall, people showed a substantial amount of stress and anxiety (on a scale from 0 to 100, the mean for stress and anxiety was 56% at T2, and 49% at T3), and worries about their finances (same scale; 52% at T2 and 48% at T3).

For stress and anxiety, a 2 (person: self/other) ∗ 2 (time point: T2/3) repeated-measures ANOVA showed a significant main effect of person ($F_{1,431} = 107.01$, $p < 0.001$, $\eta_p^2 = 0.4316$), which suggests an

**Table 2.** Results of the regression analyses, in which we tried to predict the number of physical contacts during COVID from risk perception measures, controlling for number of contacts before COVID.

| t-stat/p-value | Get COVID absolute | Get COVID relative | Infect Others absolute | Infect Others relative | Severe Symptoms absolute | Severe Symptoms relative | contacts normal |
|---|---|---|---|---|---|---|---|
| contacts COVID family | 0.99761/0.31904 | 2.626/0.0089 | 1.1296/0.25927 | 1.0303/0.30344 | −2.8964/0.00397 | 0.1668/0.86761 | 17.133/<0.001 |
| contacts COVID friends | −0.0354/0.97177 | 0.50616/0.51301 | 0.05818/0.95363 | 1.3411/0.1806 | 0.85741/0.3917 | 1.2654/0.20641 | 4.8257/<0.001 |
| contacts COVID colleagues | 0.43418/0.66438 | 1.2807/0.20099 | 0.14622/0.88382 | 0.13968/0.88898 | 0.21475/0.83007 | 0.46187/0.64441 | 4.4804/<0.001 |
| contacts COVID recreation | 0.71149/0.47717 | 1.3014/0.19381 | 0.80353/0.42212 | 0.46249/0.64397 | 0.92356/0.35624 | 1.2175/0.22409 | 4.3567/<0.001 |
| contacts COVID travel | 0.24762/0.80455 | 1.2133/0.22568 | 0.88684/0.37567 | 0.20748/0.83573 | 1.4231/0.15546 | 1.2215/0.22258 | 8.1577/<0.001 |
| contacts COVID public chores | 0.73008/0.46574 | 0.82314/0.41089 | −0.5073/0.61225 | −0.5900/0.55548 | 1.0484/0.29507 | 1.5094/0.13194 | 8.2732/<0.001 |
| contacts COVID mean | 0.48243/0.62975 | 1.5383/0.12471 | 0.44052/0.65978 | 0.92224/0.35693 | −0.07803/0.93784 | 0.74961/0.45391 | 6.8768/<0.001 |

optimism bias, and a significant main effect of time point ($F_{1,431} = 31.65$, $p < 0.001$, $\eta_p^2 = 0.2476$), which suggests that risk perception decreased over time. The interaction between person and time ($F_{1,431} = 0.20$, $p = 0.9675$, $\eta_p^2 = 0.001$) was not significant, suggesting that the optimism bias did not change over time. For financial worries, a similar picture emerged: a 2 (person: self/other) * 2 (time point: T2/3) repeated-measures ANOVA showed a significant main effect of person ($F_{1,431} = 222.97$, $p < 0.001$, $\eta_p^2 = 0.6461$), which suggests an optimism bias, and a significant main effect of time point ($F_{1,431} = 17.92$, $p < 0.001$, $\eta_p^2 = 0.1035$), which suggests that risk perception decreased over time. The interaction between person and time ($F_{1,431} = 1.34$, $p = 0.2482$, $\eta_p^2 = 0.0031$) was not significant, suggesting that the optimism bias did not change over time. Thus, for both stress/anxiety and financial worries, people showed an optimism bias in that they were less affected than others, and an overall reduction; as with all previous optimism bias questions, there was no change over time.

For stress and anxiety, absolute risk perception for the three COVID-19-related questions (*Get COVID*, *Infect Others*, *Severe Symptoms*) correlated positively with absolute stress and anxiety (all $p$-values $< 0.001$; for all values in this section, see electronic supplementary material, appendix K), and negatively with optimism bias for stress and anxiety (all $p$-values $< 0.0034$). Thus, the higher one believes the risk to be for oneself to suffer from one of those three risks, the higher one's absolute stress and anxiety were, and the lower one's stress and anxiety was relative to other people. Relative risk perception for *Get COVID* and *Infect Others* did not correlate with either absolute or relative stress and anxiety, but for *Severe Symptoms* relative risk perception correlated negatively with absolute stress and anxiety ($p = 0.0078$), and positively with relative stress and anxiety ($p < 0.001$). Thus, the higher one's optimism bias for *Severe Symptoms* was, the lower one's stress and anxiety was, and the higher one's optimism bias for stress and anxiety was. For financial worries, only absolute risk perception for *Infect Others* and *Severe Symptoms* correlated significantly, in the same pattern as for stress and anxiety: the higher one's perceived risk for infecting others and getting severe symptoms was, the higher one's absolute financial worries were and the lower one's optimism bias about financial worries was. While we cannot establish any causal relationships here, it seems clear that there is some sort of relationship between stress and anxiety, and financial worries on the one hand, and risk perception about COVID-19 on the other hand.

## 3.4. Replication

In addition to the longitudinal data from Sample 1, we also collected cross-sectional data from new samples at each time point (Samples 2–4). We used these additional samples to test which of our findings reported above replicate in new samples. To do so, we selected the most important findings from our study and preregistered our replication plan (https://osf.io/ukz8n).

We summarize the overall results here, see electronic supplementary material, appendix L for all details. Almost all main findings reported above replicated in samples 2–4: people still showed a comparative optimism bias for *Get COVID* and *Infect Others*, and none of the optimism biases changed over time; one difference here is that in our replication we found a small significant optimism bias for *Severe Symptoms*. All exploratory findings relating to control and optimism bias replicated. People were still largely aware of known risk factors, with some minor changes. Samples 2–4 also showed ceiling effects about physical distancing and abiding by hygiene recommendations, but this was less pronounced for Sample 4 at T3. Thus, in summary, the results reported above replicate almost identically in new samples.

# 4. Discussion

To successfully avert the COVID-19 pandemic, citizens need to abide by protective measures, such as reducing physical contacts, disinfecting hands and wearing masks. Their willingness to adhere to such guidelines might depend on their risk perception of the situation. During the first months of the pandemic in Western countries (i.e. March to May 2020), we asked people in Germany, the UK and the USA to rate perceived risks for COVID-19, engagement with protective measures, and various related factors. Overall, several main findings emerged:

First, people showed a strong comparative optimism bias with respect to getting infected with COVID-19 and for infecting others (if infected themselves). This bias was not present for the probability of getting severe symptoms in our longitudinal sample, but there was a small but statistically significant effect in the cross-sectional replication sample. Comparative optimism has been

found in several other studies: for example, Wise *et al*. [29], Raude *et al*. [31] and Globig *et al*. [32] all found evidence for comparative optimism for getting infected with COVID-19. As far as we are aware, there is currently no study that has assessed optimism bias with respect to infecting other people with COVID-19. In that light, it is particularly noteworthy that the optimism bias for *Infect Others* is the strongest bias related to COVID-19 in our study.

Second, the optimism bias scores for the three COVID-19-related questions remained stable over time, even though absolute risk perception changed during the same time. There were no statistically significant changes across time for any of the three questions at any time point in both the longitudinal and the cross-sectional sample. These findings are in seeming contradiction to the findings by Wise *et al*. [29], who found that the optimism bias for getting infected with COVID-19 reduced during the early stage of the pandemic. There might be a simple explanation for this difference: while Wise *et al*. collected data from 11 March until 16 March, our data collection started on 16 March. Thus, the last day in the Wise *et al*. study is the first day in our study. It is possible that for our participants there was a reduction in optimism bias in the week preceding our data collection, which then remained stable thereafter. It is thus possible that had we begun data collection a week earlier we would also have found a reduction in optimism bias in the beginning.

Third, an interesting and unexpected observation is that the strength of the comparative optimism scores differed between the three questions about the personal impact of COVID-19: *Infect Others* had the largest effect, followed by *Get COVID*; *Severe Symptoms* had no or only a small comparative optimism bias. A speculative interpretation of these results might be that the strength of one's comparative optimism relates to the perceived control over the outcome: while it is possible to almost guarantee to not infect someone else if infected with COVID-19 oneself (one can stay at home alone almost all the time, always disinfect hands and always wear a mask when buying groceries), one has less control over whether one will get infected if one has to leave the house for necessary trips such as buying groceries (even when doing the same as above, one cannot control whether a stranger will cough in one's face without wearing a mask themselves), and for getting severe symptoms if infected, at the time of data collection there was no vaccine and no cure for COVID-19, such that the probability of experiencing severe symptoms if infected could not be changed. It thus seems possible that this perceived sense of agency over the outcome might have affected to what extent people showed comparative optimism. Our data allowed for testing this idea in similar questions, all of which seemed to confirm that the perceived sense of control over an action or outcome affected people's sense of control: for example, there were significant differences between control optimism questions (STD > bone fracture > influenza), the proximal something was the more optimism people showed for that item (e.g. more optimism bias for shorter compared with longer time horizons for *Get COVID*), and items which one could not change showed reduced optimism (e.g. for *Infect Others* the social context 'family' had the lowest optimism bias). In our preregistered replication, all findings about control and optimism bias replicated, such that our initial exploratory findings seem to be robust findings that replicate to novel samples. Aligned with these findings, Globig *et al*. [32] also found that people's sense of agency predicted comparative optimism bias.

Fourth, our participants seemed generally well-informed about potential causes of COVID-19-related risks, and their risk perceptions were related to information available at the time. For example, older people reported higher absolute risks of getting *Severe Symptoms* than younger people. Similarly, the worse someone's health was, the higher they rated their absolute risk perception for *Severe Symptoms*. One notable difference was that men did not report higher risks of getting *Severe Symptoms* than women; in fact, the opposite was true: men reported lower risk than women. There was no difference in optimism bias though between men and women, indicating that men rated the chance of getting *Severe Symptoms* as lower in general, not specifically for themselves. Nonetheless, overall, it seems as if our participants were aware of the relevant factors for COVID-19. Thus, although COVID-19 was a new disease, and information was sparse and changed over time, it seems as if our participants were well-informed about possible risks.

Fifth, most participants reported almost complete engagement with protective measures, such that most participants reported an almost complete reduction in physical contacts, and complete adherence to maintaining hygiene recommendations, such as washing hands frequently. This meant that there was not enough variance for us to adequately test the relationship between risk perception and adherence to protective measures with change scores. Using regression analyses, we did not find any evidence that risk perception predicted adherence to protective measures. One has to question to what extent this self-reported adherence to protective measures is true—after all, if everyone had completely adhered to protective measures, the number of infected people would have been lower. It

is thus possible that our participants overestimated the extent to which they reduced contacts and abided by hygiene, or that our sample was not representative.

Sixth, for mental health, there seems to be a clear relationship between stress, anxiety, and financial worries on the one hand, and risk perception about COVID-19 on the other hand. While we cannot establish the direction of causality, there is a link between risk perception and mental health about COVID-19-related issues. For example, being under higher risk of getting infected might increase anxiety and worry about losing one's job, but it is also likely that living in an area with high infection rates will increase both risk perception and worry about the situation. Future studies could try to disentangle these complex relationships.

Our study has strengths and weaknesses. While we collected data in three countries that are largely similar (all are Western, educated, industrialized, rich and democratic [33]), but had different governmental reactions to the pandemic, and different trajectories for the case numbers, we do not have enough participants in each country to make particularly valid claims about the differences between countries on a nuanced level (e.g. how county/state-wide regulations affected people's beliefs and behaviours). Larger samples and more frequent data collection in each country would have been useful to study the effects of local regulations. Furthermore, our samples are not representative (with respect to age and gender; we did not measure socio-economic status, political affiliation, etc.), and it would be advisable for future studies to use representative samples from different cultures to draw stronger conclusions about how well findings generalize within and between populations. At the time of the first data collection, it was also not known widely that wearing facemasks would play such a pivotal role in reducing transmission, such that we did not ask people about wearing masks at the first two time points (but we added it at T3). Considering how contentious mask wearing became in the proceeding months in some places, it would have been interesting to see how its use in our samples changed over time. While in hindsight it seems obvious that covering one's airways is a sensible way to stop an airborne disease, when we began data collection there was a much larger focus on hand washing and hand disinfecting, which is why even at T3 we considered face masks an additional item (as part of hygiene measures), rather than a separate question. On the plus-side, at each time point, we collected a rich dataset with many questions for exploratory analyses and additional independent datasets that we used to replicate our main findings. This ability to replicate our main findings is particularly crucial considering that worldwide pandemics do not occur particularly frequently, such that replications cannot be done at will. Due to our additional datasets, we were able to show that almost all of our exploratory findings replicated.

Our study provides a detailed description of risk perception about COVID-19, as well as potential causes and consequences during the early stages of the pandemic. We hope that these results further our understanding of rapidly evolving risk perceptions in the real world, and that these data may provide valuable information to assist policy implementation in the case of future pandemics.

Ethics. The study was conducted in accordance with the Declaration of Helsinki. All participants gave informed consent before completing the online questionnaire. The study asked no potentially triggering questions and was clearly labelled as a study about the Coronavirus. The George Washington University Institutional Review Board has approved online data collection of this form (IRB#NCR191133).

Data accessibility. Data and relevant code for this research work are stored in GitHub: https://github.com/dnhi-lab/COVID-risk and have been archived within the Zenodo repository: https://doi.org/10.5281/zenodo.5570390.

Authors' contributions. B.J.K.-S. collected the data, helped preprocess the data, analysed the data, administered the project, created some of the data visualizations and wrote the original draft of the manuscript. L.M.D. helped collect the data and created most of the data visualizations. C.W.K. collected the data, preprocessed the data, acquired funding, supervised the project and helped writing the original draft of the manuscript. All authors conceived of the study, selected the questions and reviewed and edited the manuscript. All authors gave final approval for publication.

Competing interests. We declare we have no competing interests.

Funding. This work was supported by the Emmy Noether Research Group grant (392443797) from the German Research Foundation (DFG).

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
