## [Peer Review File · Royal Society Open Science]

Review History

RSOS-210904.R0 (Original submission)

Review form: Reviewer 1 (Toby Wise)

Is the manuscript scientifically sound in its present form?

Yes

Are the interpretations and conclusions justified by the results?

Yes

Is the language acceptable?

Yes

Do you have any ethical concerns with this paper?

No

Have you any concerns about statistical analyses in this paper?

No

Recommendation?

Accept with minor revision (please list in comments)

Comments to the Author(s)

This paper by Kuper-Smith and colleagues presents a thorough investigation into risk perception, and the ways in which it can be biased, during the COVID-19 pandemic. The pandemic presents a unique window into how optimistic biases in risk perception (i.e. underestimating one's risk relative to others') develop in the face of a pervasive real-world threat, and the authors have done an excellent job in understanding where this bias emerges. Briefly, they demonstrate a strong optimism bias in perceptions of infection likelihood and the likelihood of transmitting infection, but not in the expected severity of an infection, suggesting that optimism depended on the perceived control over an outcome. Notably, this bias appeared to remain over 2 months, and was higher in subjects from the US compared to UK and Germany. The study is especially strong in its use of an international sample and longitudinal design, with additional replications conducted in cross-sectional samples. The findings are important for our understanding how risk perception evolves in the real world.

Overall, the paper is timely, interesting, and presents a series of thorough and well-conducted studies. I have some minor comments/suggestions outlined below, but otherwise I would be very happy to see this paper accepted.

Signed,
Toby Wise

- Section 3.0 and the first part of 3.1 might be better placed in the methods as they include some quite important methodological details, and don't really mention any results.
- What did drop out look like across the 3 time points? How many subjects did all time points, and were only these subjects included in analyses?
- Page 9 paragraph 1 – the way the results of the preregistered hypothesis test are written is a little confusing. Stating that evidence was found for the 2 week time horizon makes it sound a little like the other time points were also hypothesized but no evidence was found, rather than that they just weren't preregistered (it's easy to miss that the preregistration only included the 2 week horizon). This is clarified in the next paragraph, so it might be worth just shifting this clarification to the earlier paragraph.
- Given the null findings regarding change in optimism bias over time, it would be nice to see some Bayesian analyses to provide a bit more information regarding the strength of the evidence for the null hypothesis (i.e. presenting Bayes factors or something similar). This isn't a major issue though, it's up to the authors if they feel this is worth adding.
- Was any information collected about the type of media people were consuming (e.g. news, social media)? I would imagine this could have a substantial influence on whether risk perception increases or decreases
- For the intended analyses predicting change in behavior based on optimism bias, I don't know whether the skew in the data makes these entirely infeasible. First, the skew isn't too bad for the public chores variable so could probably used without too much trouble. Second, it would be worth running analyses with the others anyway and checking for violations of model assumptions to be sure (alternatively, it might be possible to split subjects into those who report 0 contacts and those who report >0, then use a logistic regression or similar). Finally (and perhaps most importantly), change scores are probably not the best dependent variable here (for the reasons described in the paper). Instead, it would be better to use a regression approach, predicting contacts at T2/T3 from optimism bias, controlling for contacts pre-Covid.
- It's not entirely clear what the person variable represents in the stress & anxiety analyses – were subjects asked about their own anxiety and that of another person?

- It would be nice to see some information about the effect of horizon. Using an average for the main analyses seems like the best approach, but it would be helpful to get a sense of how risk perception varied depending on the horizon (e.g. a figure somewhere)
- Generally, it would be helpful to have a few more of the figures from the Appendix presented in the main paper – it looks like there's plenty of space remaining and some of the figures are quite helpful in illustrating points made in the text (e.g. about the skewed distributions for the behavior variables)
- On page 9, line 268 there is a typo ("statistically" should be "statistically")

Decision letter (RSOS-210904.R0)

Dear Mr Kuper-Smith

On behalf of the Editors, we are pleased to inform you that your Manuscript RSOS-210904 "Risk perception and optimism bias during the early stages of the COVID-19 pandemic" has been accepted for publication in Royal Society Open Science subject to minor revision in accordance with the referees' reports. Please find the referees' comments along with any feedback from the Editors below my signature.

Please submit your revised manuscript and required files (see below) no later than 7 days from today's (ie 16-Sep-2021) date. Note: the ScholarOne system will 'lock' if submission of the revision is attempted 7 or more days after the deadline. If you do not think you will be able to meet this deadline please contact the editorial office immediately.

on behalf of Professor Geoff Haddock (Associate Editor) and Essi Viding (Subject Editor)
openscience@royalsociety.org

Associate Editor Comments to Author (Professor Geoff Haddock):

Thank you for submitting your paper to RSOS. I want to start by apologising for the delay in being able to offer feedback on your paper. The journal experienced difficulty securing two reviews for the paper. Instead of adding to any delay, I am prepared to make a decision based on the one received review, along with my own evaluation of the paper. Both the reviewer and I have convergent views of the research; we agree that it is timely and offers a thorough consideration of risk perception and the optimism bias in the context of Covid-19. As a result, I have elected to accept the paper for publication in RSOS, subject to the completion to some revisions, which I outline below.

The reviewer's comments are clear, and for the sake of brevity I will not repeat them here.

Needless to say, I anticipate that their comments will be addressed in a revision. In addition to the reviewer's points, there are a few other issues I'd like you to address, in order to further enhance the paper. For the sake of parsimony, I order them in relation to where they appear in the manuscript.

- Page 8 - I think it would be useful to flesh out the justification of your hypotheses in a bit more detail, in order to further enhance the clarity of your predictions.
- Page 8 - Regarding the indices that were derived, presumably they had sufficient internal consistency - please provide details. Similarly, please note in the text the extent to which the Get COVID, Infect Others, and Severe Symptoms were correlated with each other (when indices were used for the first two constructs).
- Throughout the results, there were instances where having Tables and Figures in the text would be helpful for the reader.
- Page 14 - It would be useful to explain why there are higher optimism bias scores among US respondents. This could be noted within the discussion.
- It would be useful to structure the revised manuscript using journal guidelines
- Page 6 - line 161 - I believe there is a typo.
- Page 15, line 514 - need to add "p="

I expect that these revisions should be relatively straightforward to complete. When resubmitting the paper, please provide a cover letter outlining the revisions that have been made.

Thank you for submitting the paper to RSOS, and I wish you continued success in your research activities.

Sincerely,
Professor Geoff Haddock

Reviewer comments to Author:

Reviewer: 1

Comments to the Author(s)

This paper by Kuper-Smith and colleagues presents a thorough investigation into risk perception, and the ways in which it can be biased, during the COVID-19 pandemic. The pandemic presents a unique window into how optimistic biases in risk perception (i.e. underestimating one's risk relative to others') develop in the face of a pervasive real-world threat, and the authors have done an excellent job in understanding where this bias emerges. Briefly, they demonstrate a strong optimism bias in perceptions of infection likelihood and the likelihood of transmitting infection, but not in the expected severity of an infection, suggesting that optimism depended on the perceived control over an outcome. Notably, this bias appeared to remain over 2 months, and was higher in subjects from the US compared to UK and Germany. The study is especially strong in its use of an international sample and longitudinal design, with additional replications

conducted in cross-sectional samples. The findings are important for our understanding how risk perception evolves in the real world.

Overall, the paper is timely, interesting, and presents a series of thorough and well-conducted studies. I have some minor comments/suggestions outlined below, but otherwise I would be very happy to see this paper accepted.

Signed,
Toby Wise

- Section 3.0 and the first part of 3.1 might be better placed in the methods as they include some quite important methodological details, and don't really mention any results.
- What did drop out look like across the 3 time points? How many subjects did all time points, and were only these subjects included in analyses?
- Page 9 paragraph 1 – the way the results of the preregistered hypothesis test are written is a little confusing. Stating that evidence was found for the 2 week time horizon makes it sound a little like the other time points were also hypothesized but no evidence was found, rather than that they just weren't preregistered (it's easy to miss that the preregistration only included the 2 week horizon). This is clarified in the next paragraph, so it might be worth just shifting this clarification to the earlier paragraph.
- Given the null findings regarding change in optimism bias over time, it would be nice to see some Bayesian analyses to provide a bit more information regarding the strength of the evidence for the null hypothesis (i.e. presenting Bayes factors or something similar). This isn't a major issue though, it's up to the authors if they feel this is worth adding.
- Was any information collected about the type of media people were consuming (e.g. news, social media)? I would imagine this could have a substantial influence on whether risk perception increases or decreases
- For the intended analyses predicting change in behavior based on optimism bias, I don't know whether the skew in the data makes these entirely infeasible. First, the skew isn't too bad for the public chores variable so could probably be used without too much trouble. Second, it would be worth running analyses with the others anyway and checking for violations of model assumptions to be sure (alternatively, it might be possible to split subjects into those who report 0 contacts and those who report >0, then use a logistic regression or similar). Finally (and perhaps most importantly), change scores are probably not the best dependent variable here (for the reasons described in the paper). Instead, it would be better to use a regression approach, predicting contacts at T2/T3 from optimism bias, controlling for contacts pre-Covid.
- It's not entirely clear what the person variable represents in the stress & anxiety analyses – were subjects asked about their own anxiety and that of another person?
- It would be nice to see some information about the effect of horizon. Using an average for the main analyses seems like the best approach, but it would be helpful to get a sense of how risk perception varied depending on the horizon (e.g. a figure somewhere)
- Generally, it would be helpful to have a few more of the figures from the Appendix presented in the main paper – it looks like there's plenty of space remaining and some of the figures are quite helpful in illustrating points made in the text (e.g. about the skewed distributions for the behavior variables)
- On page 9, line 268 there is a typo (“statistically” should be “statistically”)

===PREPARING YOUR MANUSCRIPT===

one version identifying all the changes that have been made (for instance, in coloured highlight, in bold text, or tracked changes);
 a 'clean' version of the new manuscript that incorporates the changes made, but does not highlight them. This version will be used for typesetting.

===PREPARING YOUR REVISION IN SCHOLARONE===

- Any electronic supplementary material (ESM).
- If you are requesting a discretionary waiver for the article processing charge, the waiver form must be included at this step.
- If you are providing image files for potential cover images, please upload these at this step, and inform the editorial office you have done so. You must hold the copyright to any image provided.
- A copy of your point-by-point response to referees and Editors. This will expedite the preparation of your proof.

- Ensure that your data access statement meets the requirements at <https://royalsociety.org/journals/authors/author-guidelines/#data>. You should ensure that you cite the dataset in your reference list. If you have deposited data etc in the Dryad repository, please only include the 'For publication' link at this stage. You should remove the 'For review' link.
- If you are requesting an article processing charge waiver, you must select the relevant waiver option (if requesting a discretionary waiver, the form should have been uploaded at Step 3 'File upload' above).
- If you have uploaded ESM files, please ensure you follow the guidance at <https://royalsociety.org/journals/authors/author-guidelines/#supplementary-material> to include a suitable title and informative caption. An example of appropriate titling and captioning may be found at https://figshare.com/articles/Table_S2_from_Is_there_a_trade-off_between_peak_performance_and_performance_breadth_across_temperatures_for_aerobic_scope_in_teleost_fishes_/3843624.

Author's Response to Decision Letter for (RSOS-210904.R0)

See Appendix A.

Decision letter (RSOS-210904.R1)

Dear Mr Kuper-Smith,

I am pleased to inform you that your manuscript entitled "Risk perception and optimism bias during the early stages of the COVID-19 pandemic" is now accepted for publication in Royal Society Open Science.

If you have not already done so, please ensure that you send to the editorial office an editable version of your accepted manuscript, and individual files for each figure and table included in

your manuscript. You can send these in a zip folder if more convenient. Failure to provide these files may delay the processing of your proof.

COVID-19 rapid publication process:

We are taking steps to expedite the publication of research relevant to the pandemic. If you wish, you can opt to have your paper published as soon as it is ready, rather than waiting for it to be published the scheduled Wednesday.

This means your paper will not be included in the weekly media round-up which the Society sends to journalists ahead of publication. However, it will still appear in the COVID-19 Publishing Collection which journalists will be directed to each week (<https://royalsocietypublishing.org/topic/special-collections/novel-coronavirus-outbreak>).

If you wish to have your paper considered for immediate publication, or to discuss further, please notify openscience_proofs@royalsociety.org and press@royalsociety.org when you respond to this email.

on behalf of Professor Geoff Haddock (Associate Editor) and Essi Viding (Subject Editor)
openscience@royalsociety.org

Appendix A

Dear Professor Haddock,

Thank you very much for your work and your positive evaluation of our work. We have addressed all comments by you and by Reviewer 1, please find our revisions below.

Best wishes,

Benjamin Kuper-Smith and Christoph Korn (on behalf of all authors of this manuscript)

Comments by the editor and reviewer 1 are in Calibri font.

Our responses to the comments are in bold and italics in Calibri font.

The text from the manuscript is in Arial font and indented (changes are in red)

Associate Editor Comments to Author (Professor Geoff Haddock)

Thank you for submitting your paper to RSOS. I want to start by apologising for the delay in being able to offer feedback on your paper. The journal experienced difficulty securing two reviews for the paper. Instead of adding to any delay, I am prepared to make a decision based on the one received review, along with my own evaluation of the paper. Both the reviewer and I have convergent views of the research; we agree that it is timely and offers a thorough consideration of risk perception and the optimism bias in the context of Covid-19. As a result, I have elected to accept the paper for publication in RSOS, subject to the completion to some revisions, which I outline below.

The reviewer's comments are clear, and for the sake of brevity I will not repeat them here. Needless to say, I anticipate that their comments will be addressed in a revision. In addition to the reviewer's points, there are a few other issues I'd like you to address, in order to further enhance the paper. For the sake of parsimony, I order them in relation to where they appear in the manuscript.

We would like to thank you very much for your insightful comments and for acting as a reviewer to avoid delays in the publication process.

- Page 8 – I think it would useful to flesh out the justification of your hypotheses in a bit more detail, in order to further enhance the clarity of your predictions.

We have added justifications after each hypothesis, as specified in the preregistration. The paragraph now reads:

This study includes preregistered analyses (<https://osf.io/89ndm>) for T2. Specifically, our three preregistered hypotheses were: First, people would show an optimism bias at T2 for the questions *Get COVID* (for the time horizon 'next 2 weeks') and *Infect Others*. ***This hypothesis rests on the empirical data we collected at T1 and had published in our initial preprint (Version 1 of (23)), and on a similar study by Wise et al (29) that also found a comparative optimism bias for getting infected with COVID-19.*** Second, these optimism biases would reduce from T1 to T2; ***again this hypothesis is based on the previous results by Wise et al (29) who found a reduction of optimism***

bias in the first week of the pandemic. Third, there would be a negative correlation between these optimism biases at T1 and the reported reduction of physical contacts at T2. This hypothesis was based on our initial empirical observations from T1, during which we found a negative correlation between the optimism bias of *Infect Others*, and the perceived necessity of abiding by best practices, such that the stronger the optimistic bias about infecting other was, the less necessary people believed it was to reduce social contacts. Although the preregistration only explicitly mentioned T2, the same logic can be extended to T3. We therefore also test these preregistered hypotheses for the data from T3. As specified in our preregistration, our cut-off for significance testing was $p < 0.005$. All analyses not explicitly labelled as preregistered hypotheses are treated as exploratory analyses; for these exploratory analyses we use a cut-off for significance testing of $p < 0.05$. The replication was also preregistered (<https://osf.io/ukz8n>).

- Page 8 – Regarding the indices that were derived, presumably they had sufficient internal consistency - please provide details. Similarly, please note in the text the extent to which the Get COVID, Infect Others, and Severe Symptoms were correlated with each other (when indices were used for the first two constructs).

We address both points by adding the following text and table at the beginning of ‘3.1. Absolute and relative risk perception’:

The indices for *Get COVID* and *Infect Others*, calculated by averaging across time horizons (*Get COVID*) and social contexts (*Infect Others*), and by averaging across all time points (T1-3) for *Get COVID*, *Infect Others*, and *Severe Symptoms*, had high internal consistency (Cronbach’s alpha for *Get COVID* absolute: 0.9183; *Get COVID* relative: 0.8941; *Infect Others* absolute: 0.7939; *Infect Others* relative: 0.7622). The absolute risk perception measures of the three questions correlated positively with the absolute risk perception measures of the other questions, as did the relative risk perception questions, and within each question there was a negative correlation between absolute and relative risk perception (see Table 1):

Rho / p-value		Get COVID		Infect Others		Severe Symptoms	
		Absolute	Relative	Absolute	Relative	Absolute	Relative
Get COVID	Absolute						
	Relative	-0.3784 / <0.001					
Infect Others	Absolute	0.2785 / <0.001	-0.0339 / 0.4816				
	Relative	0.0339 / 0.4823	0.2947 / <0.001	-0.2666 / <0.001			
Severe Symptoms	Absolute	0.3116 / <0.001	-0.1027 / 0.0329	0.2231 / <0.001	0.1633 / <0.001		
	Relative	-0.1961 / <0.001	0.2750 / <0.001	0.0492 / 0.3079	-0.0572 / 0.2356	-0.6188 / <0.001	

Table 1 – Correlations between the main variables of risk perception, separately for absolute (self) and relative (other-self) risk perception, calculated by averaging across time horizons (*Get COVID*) and social contexts (*Infect Others*), and by averaging across all time points (T1-3) for *Get COVID*, *Infect Others*, and *Severe Symptoms*

- Throughout the results, there were instances where having Tables and Figures in the text would be helpful for the reader.

We have made several changes to accommodate this suggestion (also made by Reviewer 1):

- **We moved the second figure from Appendix I (number of physical contacts during COVID) into the main text (now Fig. 9)**
- **We have moved the figure from Appendix J (how much people stick by hygiene recommendations) into the main text (now Fig. 10)**
- **We have moved the entire 'Appendix D: Figures for personal risk factors and risk perception' into the end of the section '3.2 Personal risk factors', and deleted the sentence that linked to the now non-existent appendix, and added links to the figures in the respective paragraphs**

- Page 14 – It would be useful to explain why there are higher optimism bias scores among US respondents. This could be noted within the discussion.

In the section '3.2 country' we compare absolute and relative risk perception for all 3 of our main risk perception questions (Get COVID, Infect Others, Severe Symptoms), and of these 6 comparisons, only for one (relative risk perception for Infect Others) is there a significant difference between countries. This small effect ($\eta^2_H = 0.0175$) would not be significant if correcting for multiple comparisons ($p_{\text{Bonferroni}} = 0.05/6 = 0.0083 > 0.0087$). We therefore do not believe that respondents from the US show an overall higher optimism bias. To clarify this, we have added the following sentence to the section:

Although there was an overall significant difference for relative risk perception for Infect Others, this effect has a small effect size and would not be significant if controlling for multiple comparisons ($p_{\text{Bonferroni}} = 0.005/6 = 0.0083 > 0.0087$). It does thus not seem as if there are any large and meaningful differences in risk perception between our samples in the UK, the US, and Germany.

- It would be useful to structure the revised manuscript using journal guidelines

To adhere to the journal guidelines, we:

- **Made minor changes to the abstract to get the word limit below 200**
- **Added key terms to the title page**
- **Changed the referencing style to Vancouver**

We hope that these were the changes you had in mind. Please let us know if anything else is missing, we'd be happy to change it.

- Page 6 – line 161 – I believe there is a typo.

Thank you, this has been corrected: and now reads 'Appendix A'.

- Page 15, line 514 – need to add "p="

Thank you for noticing this, we have added it.

Reviewer: 1

Comments to the Author(s)

This paper by Kuper-Smith and colleagues presents a thorough investigation into risk perception, and the ways in which it can be biased, during the COVID-19 pandemic. The pandemic presents a unique window into how optimistic biases in risk perception (i.e. underestimating one's risk relative to others') develop in the face of a pervasive real-world threat, and the authors have done an excellent job in understanding where this bias emerges. Briefly, they demonstrate a strong optimism bias in perceptions of infection likelihood and the likelihood of transmitting infection, but not in the expected severity of an infection, suggesting that optimism depended on the perceived control over an outcome. Notably, this bias appeared to remain over 2 months, and was higher in subjects from the US compared to UK and Germany. The study is especially strong in its use of an international sample and longitudinal design, with additional replications conducted in cross-sectional samples. The findings are important for our understanding how risk perception evolves in the real world.

Overall, the paper is timely, interesting, and presents a series of thorough and well-conducted studies. I have some minor comments/suggestions outlined below, but otherwise I would be very happy to see this paper accepted.

Signed,
Toby Wise

Thank you very much for reviewing this paper and for your insightful comments. It is especially gratifying to us that the first author of one of the main papers our study relates to has agreed to review our paper. Below, we address all of your comments:

- Section 3.0 and the first part of 3.1 might be better placed in the methods as they include some quite important methodological details, and don't really mention any results.

We have copied these two sections into the methods section under the heading '2.6 Overall approach and preregistered analyses', in slightly different order. The text describing the preregistrations was changed in response to the first comment of the editor (see above), but the remaining text is the same as in our previous submission.

- What did drop out look like across the 3 time points? How many subjects did all time points, and were only these subjects included in analyses?

Thank you for pointing out this omission. We have added a sentence to explicitly mention the dropouts: the relevant section (as part of '2.3 Participants') now reads (new sentences in red):

In addition to the longitudinal Sample 1, at each time point we also collected cross-sectional data from each country. While Samples 3 and 4 were intentionally collected as independent replication samples, Sample 2 was initially part of the longitudinal sample (which was reported in the first version of our preprint, see <https://psyarxiv.com/epcyb/> Version 1): not all participants that were initially in Sample 1 responded at all time points, and for those that only participated at T1 and T2 we used the data from T1 as Sample 2. **In other words, of the initial 834 participants who took part at T1, 432 (52%) took part at all 3 time points and constitute Sample 1, and**

the remaining 402 participants, who did not take part at all 3 time points, constitute Sample 2. We do not believe that there is any reason to believe that this Sample 2 will differ from the others: we never advertised our study as a longitudinal sample, and we only kept data collection open for two days to achieve a higher precision of timing.

- Page 9 paragraph 1 – the way the results of the preregistered hypothesis test are written is a little confusing. Stating that evidence was found for the 2 week time horizon makes it sound a little like the other time points were also hypothesized but no evidence was found, rather than that they just weren't preregistered (it's easy to miss that the preregistration only included the 2 week horizon). This is clarified in the next paragraph, so it might be worth just shifting this clarification to the earlier paragraph.

Thank you for this suggestion, we have shifted the clarification to the earlier paragraph (with minor textual changes so it makes sense in the new context).

- Given the null findings regarding change in optimism bias over time, it would be nice to see some Bayesian analyses to provide a bit more information regarding the strength of the evidence for the null hypothesis (i.e. presenting Bayes factors or something similar). This isn't a major issue though, it's up to the authors if they feel this is worth adding.

Given that we are using classical statistical analyses, we report effect sizes throughout the manuscript to give indications on the strength of evidence for the tested hypotheses. Since we are not conducting any formal model comparisons, we would like to refrain from adding Bayes factors.

- Was any information collected about the type of media people were consuming (e.g. news, social media)? I would imagine this could have a substantial influence on whether risk perception increases or decreases

Unfortunately, we did not collect any data about what kind of media people consumed. We have added the following sentence to the end of the relevant section:

We did not collect any data about what kind of media (e.g., news or social media) people consumed, so our data does not provide any information about how different types of media might relate to risk perception.

- For the intended analyses predicting change in behavior based on optimism bias, I don't know whether the skew in the data makes these entirely infeasible. First, the skew isn't too bad for the public chores variable so could probably be used without too much trouble. Second, it would be worth running analyses with the others anyway and checking for violations of model assumptions to be sure (alternatively, it might be possible to split subjects into those who report 0 contacts and those who report >0, then use a logistic regression or similar). Finally (and perhaps most importantly), change scores are probably not the best dependent variable here (for the reasons described in the paper). Instead, it would be better to use a regression approach, predicting contacts at T2/T3 from optimism bias, controlling for contacts pre-Covid.

Thank you for this comment. We have now added regression analyses as suggested (predicting contacts at T2 from risk perception T1, controlling for contacts pre-COVID). We did this by adding the following paragraphs and table to section '3.3. Reduction of physical contacts':

Due these floor effects, the difference ($\text{Contacts}_{\text{Normal}} - \text{Contacts}_{\text{COVID}}$) is almost identical to the pre-COVID-19 numbers (because for most participants and most contexts, zero is subtracted). Thus, the amount contacts pre-COVID-19 and the reduction of contacts during COVID-19 (i.e., the difference) show very high correlations (R s of ~ 0.7 - 0.8 for friends, colleagues, recreation, and travel). Thus, taking the difference between pre-COVID-19 and during COVID-19 does not measure the reduction of physical contacts in a meaningful way. Almost everyone in our sample reported to have reduced as much as they could. While this might be good for society (and was in part due to governmental restrictions), this lack of variability in physical reductions precluded us from assessing the relationship between risk perception and reduction of physical contacts **with change scores**.

We instead opted for using regressions to test for the relation between the number of contacts during COVID (measured at T2) to the six measures of risk perception (measured at T1), controlling for the number of physical contacts before COVID. We ran separate regressions for each social context, such that we tried to predict physical contacts in each social context (family, friends, colleagues, recreation, travel, public chores, and the average of those 6 social contexts).

t-stat / p-value	Get COVID absolute	Get COVID relative	Infect Others absolute	Infect Others relative	Severe Symptoms absolute	Severe Symptoms relative	Contacts normal
Contacts COVID Family	0.99761 / 0.31904	2.6262 / 0.0089	1.1296 / 0.25927	1.0303 / 0.30344	-2.8964 / 0.00397	0.1668 / 0.86761	17.133 / <0.001
Contacts COVID Friends	-0.0354 / 0.97177	0.50616 / 0.51301	0.05818 / 0.95363	1.3411 / 0.1806	0.85741 / 0.3917	1.2654 / 0.20641	4.8257 / <0.001
Contacts COVID Colleagues	0.43418 / 0.66438	1.2807 / 0.20099	0.14622 / 0.88382	0.13968 / 0.88898	0.21475 / 0.83007	0.46187 / 0.64441	4.4804 / <0.001
Contacts COVID Recreation	0.71149 / 0.47717	1.3014 / 0.19381	0.80353 / 0.42212	0.46249 / 0.64397	0.92356 / 0.35624	1.2175 / 0.22409	4.3567 / <0.001
Contacts COVID Travel	0.24762 / 0.80455	1.2133 / 0.22568	0.88684 / 0.37567	0.20748 / 0.83573	1.4231 / 0.15546	1.2215 / 0.22258	8.1577 / <0.001
Contacts COVID Public chores	0.73008 / 0.46574	0.82314 / 0.41089	-0.5073 / 0.61225	-0.5900 / 0.55548	1.0484 / 0.29507	1.5094 / 0.13194	8.2732 / <0.001
Contacts COVID Mean	0.48243 / 0.62975	1.5383 / 0.12471	0.44052 / 0.65978	0.92224 / 0.35693	-0.07803 / 0.93784	0.74961 / 0.45391	6.8768 / <0.001

Table 2 – Results of the regression analyses, in which we tried to predict the number of physical contacts during COVID from risk perception measures, controlling for number of contacts before COVID

Overall, results were similar across all social contexts: risk perception did not predict the number of physical contacts, but the number of physical contacts before COVID was always a significant predictor. Table 2 shows the results for predicting physical contacts at T2 from risk perception at T1 and we repeated the same analysis for predicting physical contacts at T3 from risk perception at T2, with similar results: the number of physical contacts before COVID was always a significant predictor, but for

the context Family, the significant risk perception predictors from Table 2 were no longer significant. For some of the other contexts, in the T2/T3 analysis a single risk perception measure was a significant predictor, but not in any systematic way, and only at the level of $p=0.05$, not correcting for multiple comparisons. Thus, in our sample, it does not seem as if risk perception is a significant predictor of the number of physical contacts during the pandemic.

We also changed the abstract to incorporate these new analyses, specifically this section:

5) people reported adhering closely to protective measures but these measures did not seem to be related to risk perception

We also adapted the discussion accordingly:

Fifth, most participants reported almost complete engagement with protective measures, such that most participants reported an almost complete reduction in physical contacts, and complete adherence to maintaining hygiene recommendations, such as washing hands frequently. This meant that there was not enough variance for us to adequately test the relationship between risk perception and adherence to protective measures **with change scores. Using regression analyses, we did not find any evidence that risk perception predicted adherence to protective measures.** One has to question to what extent this self-reported adherence to protective measures is true – after all, if everyone had completely adhered to protective measures, the number of infected people would have been lower. It is thus possible that our participants overestimated the extent to which they reduced contacts and abided by hygiene, or that our sample was not representative.

- It's not entirely clear what the person variable represents in the stress & anxiety analyses – were subjects asked about their own anxiety and that of another person?

Yes, this question was set up the same way that most other questions were: rate separately how much stress/anxiety you felt, and then the same for someone similar to you. To make this clearer, we have added the following sentence in the beginning of the Mental Health section:

As with most other questions, we asked participants to rate these questions both for themselves and for someone similar to them i.e., how much stress and anxiety/financial worry they felt themselves, and how much stress and anxiety/financial worry someone similar to themselves felt.

- It would be nice to see some information about the effect of horizon. Using an average for the main analyses seems like the best approach, but it would be helpful to get a sense of how risk perception varied depending on the horizon (e.g. a figure somewhere)

We have added an ANOVA to compare time horizons for both absolute and relative risk perception in the first paragraph of the Get COVID section, which reads:

Participants in our study believed that there was a substantial risk that they would get infected with COVID-19 (**Figure 2** shows an overview of the three risk perception questions). The mean score for *Get COVID* averaged across the four time horizons for 'self' was 49% at T1, 46% at T2, and 35% at T3. For the time horizon 'lifetime', the mean score was always above 50%, indicating that participants thought they were more likely than not to get infected with COVID-19 during their lifetimes (note: our data

was collected many months before vaccinations about COVID-19 were developed and discussed publicly). While absolute risk perception (averaged across T1-3) increased with longer time horizons ($F(3, 1724) = 149.04, p < 0.001, \eta_p^2 = 0.1708$), relative risk perception (averaged across T1-3) slightly reduced with longer time horizons ($F(3, 1724) = 8.57, p < 0.001, \eta_p^2 = 0.0145$; see **Appendix C**).

We also added raincloud plots of absolute and relative risk perception for Get COVID to the Appendix C.

Similarly, we added an ANOVA to compare different social contexts for Infect Others, again for absolute and relative risk perception to the relevant section:

Participants believed that there was a substantial risk that they would infect someone else with COVID-19 (if they themselves were infected): the mean risk for *Infect Others* averaged across all social contexts was always estimated to be above 20% (T1: 39%; T2: 26%; T3: 25%). There were significant differences between the different social contexts for *Infect Others* for both absolute risk perception (averaged across T1-3; $F(5, 2586) = 218.45, p < 0.001, \eta_p^2 = 0.2290$), and for relative risk perception (averaged across T1-3; $F(5, 2586) = 28.1, p < 0.001, \eta_p^2 = 0.0490$; see **Appendix C**).

We also added figures for absolute and relative risk perception for each social context of Infect Others to Appendix C.

- Generally, it would be helpful to have a few more of the figures from the Appendix presented in the main paper – it looks like there's plenty of space remaining and some of the figures are quite helpful in illustrating points made in the text (e.g. about the skewed distributions for the behavior variables)

This has been incorporated above to a similar comment from the editor.

- On page 9, line 268 there is a typo (“statistically” should be “statistically”)

Thank you, this has been corrected.

Other (this is not the reviewer's but ours):

We found an error in Fig. 2 from the Appendix, where the density functions had been filtered incorrectly, such that some density functions for the variables were swapped. This has been corrected.